# Roles of metal ions in the selective inhibition of oncogenic variants of isocitrate dehydrogenase 1

Shuang Liu [1,2], Martine I. Abboud [1,3,4], Tobias John [1,4], Victor Mikhailov[1], Ingvild Hvinden [1], John Walsby-Tickle [1], Xiao Liu[1], Ilaria Pettinati [1], Tom Cadoux-Hudson[1], James S. O. McCullagh [1] & Christopher J. Schofield [1✉]

Cancer linked isocitrate dehydrogenase (IDH) 1 variants, notably R132H IDH1, manifest a 'gain-of-function' to reduce 2-oxoglutarate to 2-hydroxyglutarate. High-throughput screens have enabled clinically useful R132H IDH1 inhibitors, mostly allosteric binders at the dimer interface. We report investigations on roles of divalent metal ions in IDH substrate and inhibitor binding that rationalise this observation. $Mg^{2+}/Mn^{2+}$ ions enhance substrate binding to wt IDH1 and R132H IDH1, but with the former manifesting lower $Mg^{2+}/Mn^{2+}$ $K_M$s. The isocitrate-$Mg^{2+}$ complex is the preferred wt IDH1 substrate; with R132H IDH1, separate and weaker binding of 2-oxoglutarate and $Mg^{2+}$ is preferred. Binding of R132H IDH1 inhibitors at the dimer interface weakens binding of active site $Mg^{2+}$ complexes; their potency is affected by the $Mg^{2+}$ concentration. Inhibitor selectivity for R132H IDH1 over wt IDH1 substantially arises from different stabilities of wt and R132H IDH1 substrate-$Mg^{2+}$ complexes. The results reveal the importance of substrate-metal ion complexes in wt and R132H IDH1 catalysis and the basis for selective R132H IDH1 inhibition. Further studies on roles of metal ion complexes in TCA cycle and related metabolism, including from an evolutionary perspective, are of interest.

[1] Chemistry Research Laboratory, Department of Chemistry and the Ineos Oxford Institute for Antimicrobial Research, University of Oxford, 12 Mansfield Road, Oxford OX1 3TA, UK. [2] Present address: Broad Institute of MIT and Harvard, 415 Main Street, Cambridge, MA 02142, USA. [3] Present address: Department of Natural Sciences, Lebanese American University, Byblos/Beirut, Lebanon. [4] These authors contributed equally: Martine I. Abboud and Tobias John. ✉email: christopher.schofield@chem.ox.ac.uk

Changes in the small-molecule composition of cancer cells compared to normal cells correlate with mutations in genes encoding for tricarboxylic acid (TCA) cycle and related enzymes[1]. These include succinate dehydrogenase, fumarate hydratase and malate dehydrogenase, the genes for which undergo germline loss-of-function mutations associated with rare cancers such as paraganglioma and leiomyoma[2,3]. By contrast, mutations in isocitrate dehydrogenase 1/2 (*IDH1/2*) genes are somatic and are found in >80% of low-grade glioma[4] and 20—30% of acute myeloid leukaemia (AML) cases[5] (Supplementary Fig. S1).

There are three human IDH isoforms: IDH1 localises to the cytoplasm, whereas IDH2 and IDH3 localise to the mitochondria. IDH3 catalyses the irreversible oxidative decarboxylation of isocitrate to 2-oxoglutarate (2OG) in the TCA cycle, using the cofactor $NAD^+$; IDH1 and IDH2 reversibly catalyse the same reaction using $NADP^+$. Cancer-linked IDH1/2 variants have reduced wild-type (wt) activity, but gain the ability to convert 2OG to the 'oncometabolite' 2-hydroxyglutarate (2HG) (Fig. 1a), resulting in substantially elevated 2HG levels in tumours bearing

*IDH* mutations[6]. Elevated 2HG levels are proposed to inhibit enzymes, including 2OG-oxygenases, e.g. lysine histone demethylases (JmjC KDMs) and ten-eleven translocation enzymes, leading to epigenetic alternations and tumorigenesis[6]. Cancer-linked substitutions at R132 of IDH1 (to H, C, S, L, G and V) involve shorter and less basic residues than arginine, consistent with weakened binding of isocitrate (and maybe $CO_2$/bicarbonate) compared to wt IDH1, whilst still enabling 2OG binding[7]. Nearly all reported cancer-linked *IDH* gene mutations are heterozygous[4,6,8]; both the mutant homodimer and the wt/ mutant heterodimer of IDH1 can generate 2HG[9,10]. wt IDH1 and R132H IDH1 (which represents 80% of all *IDH* mutations[11]) have conserved homodimeric folds with α-helices at their dimer interface (Fig. 1b)[12,13].

Following the identification of *IDH1* mutations in glioblastoma by DNA sequencing[14], drug discovery programmes targeting mutant IDH1/2 have produced inhibitors in clinical use and development[15]. Historically, it is more likely for enzyme inhibitors to bind competitively at the same site as a (co)substrate, as supported by a review of 85 Food and Drug Administration-approved

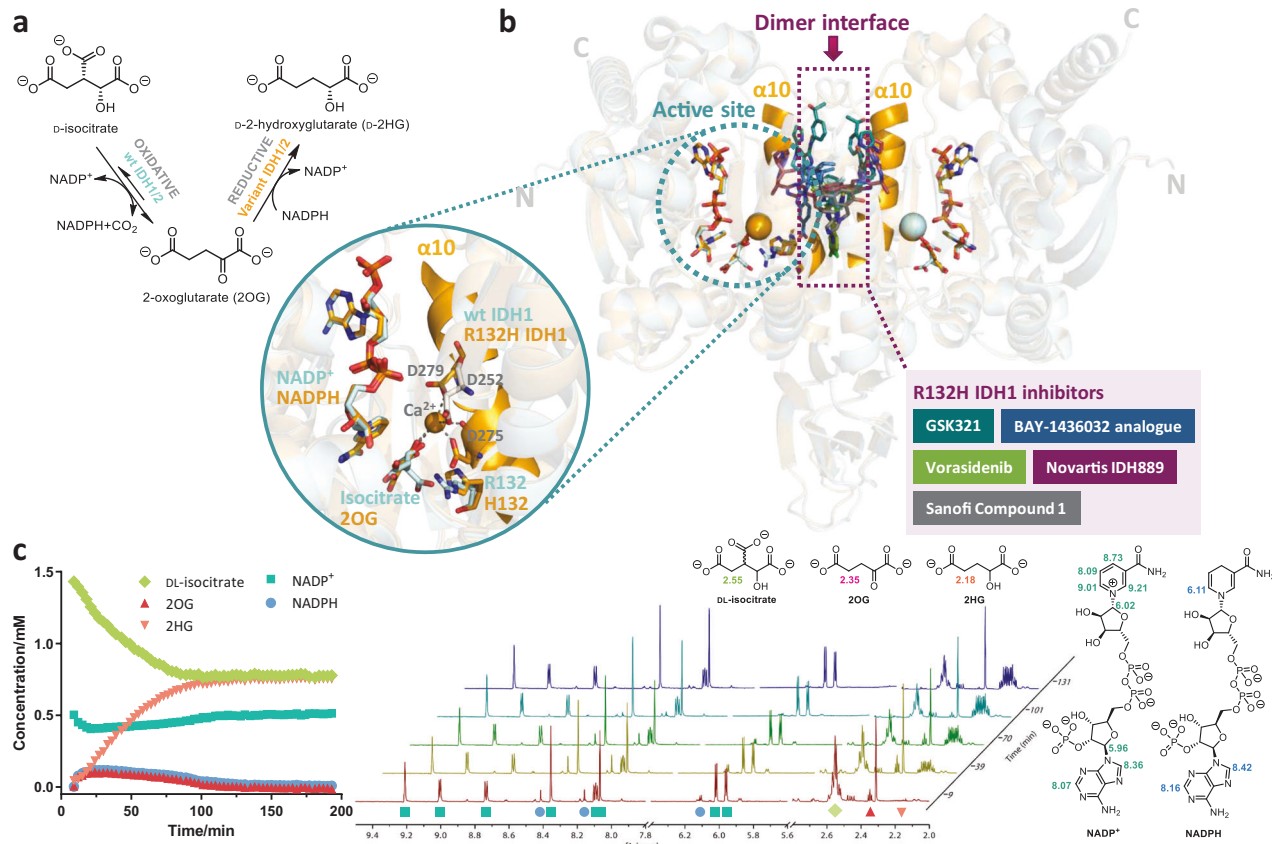

**Fig. 1 Reactions catalysed by wild-type and cancer cell harbouring IDH1/2 variants. a** Wild-type (wt) IDH1/2 catalyses the $NADP^+$-dependent oxidation of D-isocitrate to 2OG; cancer-associated IDH1/2 variants additionally catalyse NADPH-dependent 2OG reduction to D-2HG. **b** *IDH inhibitors bind at the dimer interface.* Active site and overall views of superimposed structures of homodimeric wt IDH1 (pale cyan, PDB 1T0L)[12] and R132H IDH1 (pale orange/ orange, PDB 3INM)[13]. The active site contains the cofactor ($NADP^+$ for wt IDH1; NADPH for R132H IDH1), the substrate (isocitrate for wt IDH1; 2OG for R132H IDH1) and inhibitory $Ca^{2+}$ positioned to coordinate to the substrate; $Mg^{2+}$, which is required for catalysis, likely binds in a related manner. The dimer interface where potent R132H IDH1 inhibitors bind is indicated by a purple dotted box. The active site and the dimer interface are linked through a proposed 'regulatory' α10 (orange, residues N271–G286); the metal ion and allosteric inhibitors bind on opposite faces of α10 (D275 and D279 from which are involved in metal chelation). Five R132H IDH1 allosteric inhibitor structures superimposed at the dimer interface: GSK321 (teal, PDB 5DE1)[17], BAY-1436032 analogue (navy, PDB 5LGE)[18], Vorasidenib (green, PDB 6ADG)[58], Novartis IDH889 (purple, PDB 5TQH)[59] and Sanofi 1 (dark grey, PDB 4UMX)[27]. **c** *Conversion of D-isocitrate and $NADP^+$ catalysed by R132H IDH1*, as monitored by $^1H$ NMR (700 MHz) spectroscopy. Isocitrate is converted to 2HG via 2OG. Conditions: 50 mM Tris-$D_{11}$-HCl, pH 7.5 in 90% $H_2O$/10% $D_2O$ (v/v), 2 μM R132H IDH1, 1.5 mM D-isocitrate, 500 μM $NADP^+$ and 10 mM $MgCl_2$. Initially, $NADP^+$ (teal) is converted to NADPH (blue) during the conversion of isocitrate (yellow-green) to 2OG (pink). As R132H IDH1 also catalyses the conversion of 2OG (pink) to 2HG (orange), NADPH is consumed to regenerate $NADP^+$.

drugs between 2001 and 2004, of which 80% are active site-binding substrate competitors[16]. Despite being apparently structurally diverse and selective for R132H over wt IDH1, potent R132H IDH1 inhibitors, e.g. BAY-1436032 (Bayer), GSK321 (GSK), IDH305 (Novartis) and ML309 (Agios) (half-maximal inhibitory concentration ($IC_{50}$s) < 100 nM) arising from high-throughput screens, followed by optimisation, do not bind at the active site, but at an allosteric site at the IDH dimer interface[17–21]. This striking observation suggests undetermined processes during substrate and inhibitor binding to IDH1. Although Ivosidenib and Enasidenib, which target IDH1 and IDH2 variants, respectively, are approved for AML treatment, resistance has been identified[22]. A detailed understanding of the mechanism of inhibition of IDH inhibitors and IDH catalysis may aid in the development of improved IDH targeted therapies.

IDH1 kinetics are complex—it interconverts between open and closed forms, its turnover number decreases at high enzyme concentrations and in the dimeric form, it manifests half-site reactivity with binding of $Mg^{2+}$ and isocitrate promoting release of NADPH to give an enzyme dimer with one molecule of NADPH bound[23]. Although there are pioneering studies on the role of metal ions by wt IDH2 from porcine heart[24,25], few recent studies are reported on this topic, in particular with respect to IDH inhibition.

During mechanistic studies aimed at understanding the IDH mechanism and why independent medicinal chemistry campaigns have identified allosteric inhibitors, we observed profound differences in the metal ion dependencies of wt and R132H IDH1, in particular, that the apparent Michaelis–Menten constant ($K_M$) of $Mg^{2+}$ ions for wt IDH1 is significantly lower than for R132H IDH1. This observation motivated detailed analyses on the roles of metal ions in catalysis and inhibition by wt and R132H IDH1. It is reported that some R132H IDH1 inhibitors are competitive with the substrate 2OG[17,26] and/or $Mg^{2+}$ [27]; however, this does not explain their selectivity for IDH1 variants. Here, we provide a coherent mechanism involving an interplay between wt and R132H IDH1, their substrates, metal ions and allosteric inhibitors that rationalises inhibitor selectivity for R132H over wt IDH1, despite inhibitor binding at a site remote from the active site substitution. The results reveal different roles of metal ions in isocitrate and 2OG binding to wt and R132H, respectively, and have implications for the development of improved IDH variant inhibitors and the interpretation of in vivo IDH inhibition results. They suggest that further studies on the roles of metal ion complexes in the evolution of metabolism and other metal-mediated biological systems are worthwhile.

## Results

### Differential effects of $Mg^{2+}/Mn^{2+}$ ions on isocitrate oxidation and 2OG reduction

In initial kinetic studies measuring the change in absorbance of NADPH at 340 nm to evaluate the (co)substrate preferences of recombinant wt and R132H IDH1, both proteins were ethylenediaminetetraacetic acid (EDTA)-treated to remove potential copurifying divalent metals. In accord with prior reports[7,13,27,28], under our standard assay conditions with 10 mM MgCl₂, R132H IDH1 ($K_M$ = 257 μM) has a lower affinity for isocitrate (DL-isocitrate was used except where stated) than wt IDH1 ($K_M$ = 38 μM) (Table 1), consistent with the reduced ability of R132H IDH1 to oxidise isocitrate[13]. R132H IDH1 has a high 2OG $K_M$ of 652 μM and the $k_{cat}/K_M$ of R132H IDH1 for 2OG reduction is $9.4 \times 10^{-4}$ s$^{-1}$ μM$^{-1}$, which is >1000 times lower than that of wt IDH1 for isocitrate oxidation (1.3 s$^{-1}$ μM$^{-1}$); however, this is only slightly over twofold higher than that for R132H IDH1 catalysed isocitrate oxidation ($3.9 \times 10^{-4}$ s$^{-1}$ μM$^{-1}$) (Table 1). Because the conversion of isocitrate to 2HG occurs

without net NADP$^+$ consumption, it is not possible to readily monitor the complete reaction by ultraviolet (UV) absorbance, hence nuclear magnetic resonance (NMR) was employed to validate the non-redox R132H IDH1 catalysed conversion of isocitrate to 2HG via 2OG (Fig. 1c). These observations show that, at least in isolated form, R132H IDH1 catalyses the conversions of isocitrate to 2OG and of 2OG to 2HG with similar efficiency, a finding differing from a prior study, which reported little activity of R132H IDH1 in isocitrate conversion compared to 2OG conversion by fluorescence-based assays[27]. As observed by [1]H NMR spectroscopy, only half of the racemic DL-isocitrate was converted by R132H IDH1 (Fig. 1c). Similarly for wt IDH1, D-isocitrate was selectively converted to 2OG from a mixture of DL-isocitrate; L-isocitrate was observed to be neither a substrate nor an inhibitor of wt IDH1 (Supplementary Fig. S4).

Reported studies[7,13,25,27–30] on human IDH1 lack a comprehensive characterisation of the roles of metal ions in the oxidative and reductive reactions of wt and variant IDH1s. We observed that the apparent MgCl₂ $K_M$ was ~120-fold higher for R132H IDH1 catalysed 2OG reduction (4.4 mM) than for wt IDH1 isocitrate oxidation (36 μM) (Table 1). In the presence of 150 mM NaCl, i.e. a physiologically relevant ionic strength, the MgCl₂ $K_M$ for R132H IDH1 catalysed 2OG reduction ($K_M$ = 6.7 mM, Supplementary Fig. S2m) remained significantly (70-fold) higher than that for wt IDH1 isocitrate oxidation ($K_M$ = 94 μM, Supplementary Fig. S2d). Since $Mn^{2+}$ ions are important in human metabolism[31] and are commonly used as a surrogate for $Mg^{2+}$ due to their similar properties[32,33], we substituted MnCl₂ for MgCl₂; we observed lower $K_M$s, with a 210-fold difference between R132H IDH1 ($K_M$ = 422 μM) and wt IDH1 ($K_M$ = 2.0 μM), and higher $k_{cat}$ values, for both reactions with MnCl₂ (Table 1).

### Effects of metal ions on activity, inhibition and stabilities of wt and R132H IDH1

Overall, most of our kinetic data agree with prior reports[7,13,25,27,28], although they highlight the ability of R132H to catalyse the non-redox conversion of isocitrate to 2HG and the differential effects of $Mg^{2+}/Mn^{2+}$ ions on the oxidative wt and reductive R132H IDH1 catalysed reactions. To explore the latter, we examined the effects of a range of divalent and monovalent metal ions on wt and R132H IDH1 catalysis using the UV-absorbance assay. Catalysis by wt and R132H IDH1 was only promoted by $Mg^{2+}$ and $Mn^{2+}$ (to similar levels at 10 mM) (Supplementary Fig. S5a, b). In the presence of $Mg^{2+}$ (10 mM), the metal ion inhibition profiles of wt and R132H differ, with the alkaline earth metals, $Ca^{2+}$, $Sr^{2+}$ and $Ba^{2+}$, inhibiting both wt and R132H IDH1, but to different extents (Supplementary Fig. S5c, d). $Ca^{2+}$ and $Sr^{2+}$ are established IDH inhibitors having been shown to compete with $Mn^{2+}$ binding in studies on porcine heart IDH2[25]; inhibition of R132H IDH1 by $Ca^{2+}$ was validated by [1]H NMR (Supplementary Fig. S6). Other divalent metal ions, including $Co^{2+}$, $Ni^{2+}$, $Zn^{2+}$ and $Cd^{2+}$, are also inhibitory (Supplementary Fig. S5c, d). Monovalent metal ions (Li$^+$, Na$^+$, K$^+$ and Cs$^+$) had no effect on wt IDH1 catalysis under our conditions, although Na$^+$ inhibited R132H IDH1 (Supplementary Fig. S5d).

We studied metal ion binding to wt and R132H IDH1 using differential scanning fluorimetry (DSF). Consistent with reported values measured by circular dichroism (wt IDH1 $T_m$ 49.1 °C; R132H IDH1 $T_m$ 49.7 °C)[34], the thermal stabilities of wt IDH1 ($T_m$ 52.0 °C) and R132H IDH1 ($T_m$ 50.8 °C) are similar as measured by DSF. wt IDH1 manifests clear thermal stabilisation by the addition of $Ca^{2+}$ (3.2 °C) and $Mn^{2+}$ (2.9 °C), with $Mg^{2+}$ having a less stabilising effect (Supplementary Fig. S5e). By contrast, $Mg^{2+}$, $Mn^{2+}$ and $Ca^{2+}$ destabilise R132H IDH1, with

**Table 1 Effects of magnesium and manganese ions on kinetic parameters for the oxidative and reductive reactions of wt and R132H IDH1, respectively.**

| Oxidative reaction (isocitrate to 2OG) with $Mg^{2+}$ | | | | Reductive reaction (2OG to isocitrate by wt, to 2HG by R132H) with $Mg^{2+}$ | | | |
|---|---|---|---|---|---|---|---|
| | | wt | R132H | | | wt | R132H |
| Isocitrate | $K_M$ (µM) | 38 ± 6 | 257 ± 50 | 2OG | $K_M$ (µM) | $(1.1 ± 0.3) × 10^3$ | 652 ± 116 |
| | $V_{max}$ (nM s$^{-1}$) | 216 ± 8 | 36 ± 2 | | $V_{max}$ (nM s$^{-1}$) | 200 ± 16 | 236 ± 10 |
| | $k_{cat}/K_M$ (s$^{-1}$ µM$^{-1}$) | 1.3 ± 0.3 | $(3.9 ± 1.1) × 10^{-4}$ | | $k_{cat}/K_M$ (s$^{-1}$ µM$^{-1}$) | $(1.9 ± 0.6) × 10^{-3}$ | $(9.4 ± 2.7) × 10^{-4}$ |
| NADP$^+$ | $K_M$ (µM) | 27 ± 2 | 10 ± 1 | NADPH | $K_M$ (µM) | 115 ± 19 | 15 ± 2 |
| | $V_{max}$ (nM s$^{-1}$) | 246 ± 4 | 40 ± 1 | | $V_{max}$ (nM s$^{-1}$) | 221 ± 15 | 276 ± 10 |
| | $k_{cat}/K_M$ (s$^{-1}$ µM$^{-1}$) | 1.9 ± 0.3 | $(1.0 ± 0.2) × 10^{-2}$ | | $k_{cat}/K_M$ (s$^{-1}$ µM$^{-1}$) | $(1.8 ± 0.4) × 10^{-2}$ | $(4.1 ± 1.0) × 10^{-2}$ |
| MgCl$_2$ | $K_M$ (µM) | 36 ± 4 | 171 ± 23 | MgCl$_2$ | $K_M$ (µM) | a | $(4.4 ± 0.7) × 10^3$ |
| | $V_{max}$ (nM s$^{-1}$) | 214 ± 5 | 45 ± 2 | | $V_{max}$ (nM s$^{-1}$) | | 224 ± 10 |
| | $k_{cat}/K_M$ (s$^{-1}$ µM$^{-1}$) | 1.4 ± 0.3 | $(5.8 ± 1.4) × 10^{-4}$ | | $k_{cat}/K_M$ (s$^{-1}$ µM$^{-1}$) | | $(1.4 ± 0.4) × 10^{-4}$ |
| | $k_{cat}$ (s$^{-1}$) | 50 ± 4 | 0.10 ± 0.01 | | $k_{cat}$ (s$^{-1}$) | 2.1 ± 0.1 | 0.61 ± 0.07 |

| Oxidative reaction (isocitrate to 2OG) with $Mn^{2+}$ | | | | Reductive reaction (2OG to isocitrate by wt, to 2HG by R132H) with $Mn^{2+}$ | | | |
|---|---|---|---|---|---|---|---|
| | | wt | R132H | | | wt | R132H |
| Isocitrate | $K_M$ (µM) | 35 ± 4 | 219 ± 35 | 2OG | $K_M$ (µM) | b | 175 ± 26 |
| | $V_{max}$ (nM s$^{-1}$) | 313 ± 8 | 355 ± 22 | | $V_{max}$ (nM s$^{-1}$) | | 390 ± 16 |
| | $k_{cat}/K_M$ (s$^{-1}$ µM$^{-1}$) | 2.3 ± 0.6 | $(3.6 ± 0.9) × 10^{-3}$ | | $k_{cat}/K_M$ (s$^{-1}$ µM$^{-1}$) | | $(5.9 ± 1.2) × 10^{-3}$ |
| NADP$^+$ | $K_M$ (µM) | 27 ± 3 | 80 ± 8 | NADPH | $K_M$ (µM) | | 32 ± 5 |
| | $V_{max}$ (nM s$^{-1}$) | 358 ± 7 | 302 ± 9 | | $V_{max}$ (nM s$^{-1}$) | | 433 ± 18 |
| | $k_{cat}/K_M$ (s$^{-1}$ µM$^{-1}$) | 3.0 ± 0.7 | $(9.8 ± 2.0) × 10^{-3}$ | | $k_{cat}/K_M$ (s$^{-1}$ µM$^{-1}$) | | $(3.3 ± 0.7) × 10^{-2}$ |
| MnCl$_2$ | $K_M$ (µM) | 2.0 ± 0.3 | $(1.64 ± 0.01) × 10^3$ | MnCl$_2$ | $K_M$ (µM) | | 422 ± 67 |
| | $V_{max}$ (nM s$^{-1}$) | 416 ± 13 | 290 ± 12 | | $V_{max}$ (nM s$^{-1}$) | | 421 ± 18 |
| | $k_{cat}/K_M$ (s$^{-1}$ µM$^{-1}$) | 41 ± 12 | $(4.8 ± 0.5) × 10^{-4}$ | | $k_{cat}/K_M$ (s$^{-1}$ µM$^{-1}$) | | $(2.5 ± 0.5) × 10^{-3}$ |
| | $k_{cat}$ (s$^{-1}$) | 81 ± 11 | 0.78 ± 0.08 | | $k_{cat}$ (s$^{-1}$) | | 1.04 ± 0.05 |

Reaction mixtures contain 10 mM MgCl$_2$ or MnCl$_2$. See 'Methods' for assay details. NaHCO$_3$ (100 mM, ~2.3 mM CO$_2$ at pH 8)[28] was used as a CO$_2$ source for the conversion of 2OG to isocitrate by wt IDH1. Due to the detection limit, $K_M$s could not be determined accurately for <10 µM NADP$^+$ or NADPH. See Supplementary Figs. S2 and S3 for kinetic plots. Calculated $K_M$ and $V_{max}$ values are reported as mean ± SEM, n = 3 technical replicates.
aNot determined due to conversion without added MgCl$_2$.
bNot measured due to the formation of insoluble MnCO$_3$ from MnCl$_2$ and NaHCO$_3$.

Ca$^{2+}$ having the least effect (−1.2 °C) (Supplementary Fig. S5f). The combined results reveal clear differences in the way wt IDH1 and R132H IDH1 interact with divalent metal ions.

**Binding of isocitrate to wt IDH1 is substantially influenced by the presence of metal ions.** Kinetic studies on native porcine heart IDH2 led to the conclusion that the isocitrate–Mg$^{2+}$ [35] (or isocitrate–Mn$^{2+}$ [25]) complex is the preferred substrate, i.e. isocitrate forms a complex with Mg$^{2+}$ prior to binding to the enzyme, instead of isocitrate or Mg$^{2+}$ separately binding to the enzyme. We thus investigated substrate and Mg$^{2+}$ binding to human wt IDH1 and R132H IDH1 using isothermal titration calorimetry (ITC). Neither isocitrate ($K_D$ > 1 mM, Fig. 2a) nor Mg$^{2+}$ ($K_D$ > 8 mM, Fig. 2b) alone binds tightly to wt IDH1, but in the presence of MgCl$_2$ (5 mM), isocitrate binds to wt IDH1 with a low micromolar affinity ($K_D$ = 0.9 µM, Fig. 2c; $K_D$ = 0.28 µM in the presence of 150 mM NaCl, Supplementary Fig. S7a). The enhancement of isocitrate binding to wt IDH1 by Mg$^{2+}$ was supported by DSF analyses. Isocitrate alone stabilises wt IDH1 in a dose-dependent manner (4.6–16 °C, with 0.125–4 mM isocitrate) (Fig. 2d), but the addition of Mg$^{2+}$ (10 mM) substantially enhances the stabilisation (12–19 °C, with 0.125–4 mM isocitrate) (Fig. 2f). By contrast, Mg$^{2+}$ alone does not stabilise wt IDH1 over a wide concentration range (0.032–20 mM) (Fig. 2e). Similarly, enhanced binding of isocitrate to wt IDH1 by addition of Mg$^{2+}$ was observed in the presence of 150 mM NaCl (Supplementary Fig. S8a). Both the catalytically active Mn$^{2+}$ and inhibitory Ca$^{2+}$ also promote binding of isocitrate to wt IDH1, with $K_D$s of 1.30 and 1.79 µM, respectively (Supplementary Fig. S7b, c).

To investigate whether the differences in the apparent $K_D$s of isocitrate–metal ion complex to wt IDH1 (in part) reflect differences in the binding of different metal ions to isocitrate, we measured direct binding of selected metal ions to isocitrate by ITC (Fig. 2g and Supplementary Fig. S9). Under our conditions, isocitrate binding was observed for Mg$^{2+}$ ($K_D$ = 1.5 mM, Supplementary Fig. S9a) and Mn$^{2+}$, but not for the inhibitory Ca$^{2+}$, Sr$^{2+}$, and Ba$^{2+}$. Notably, no binding was observed between 2OG and Mg$^{2+}$ by ITC (Supplementary Fig. S10). Combined with the DSF studies, these observations support the proposal that the isocitrate–Mg$^{2+}$ complex is the preferred wt IDH1 substrate; however, they suggest that the 2OG–Mg$^{2+}$ complex may be less likely to be the preferred R132H IDH1 substrate.

The proposal that the isocitrate–Mg$^{2+}$ complex is the preferred wt IDH1 substrate was also studied by non-denaturing protein mass spectrometry (MS). Initial studies implied wt and R132H IDH1 predominantly exist as dimers (Supplementary Fig. S11a–d), consistent with crystallographic analyses[12,13], with both copurifying with two NADP(H) molecules for which the cofactor redox state was shown to be, at least predominantly, NADPH by NMR (Supplementary Fig. S11e–j). The observation that wt IDH1 copurifies with two NADPH molecules differs from one previous study reporting a 1:2 NADP$^+$:NADPH mixture[28], but is consistent with another employing high-performance liquid chromatography analysis of NADP(H)[23]. The differences may reflect different purification procedures. Consistent with our MS/NMR observations, DSF studies imply that NADPH is more tightly bound than NADP$^+$ to both wt and R132H IDH1 (Supplementary Fig. S8b).

The addition of D-isocitrate to wt IDH1 (wt IDH1:D-isocitrate, 1:4) revealed new peaks corresponding to the addition of one and, less abundantly, two isocitrate molecules (Fig. 2j). The addition of Mg$^{2+}$ (wt IDH1:D-isocitrate:Mg$^{2+}$, 1:4:4) significantly increases

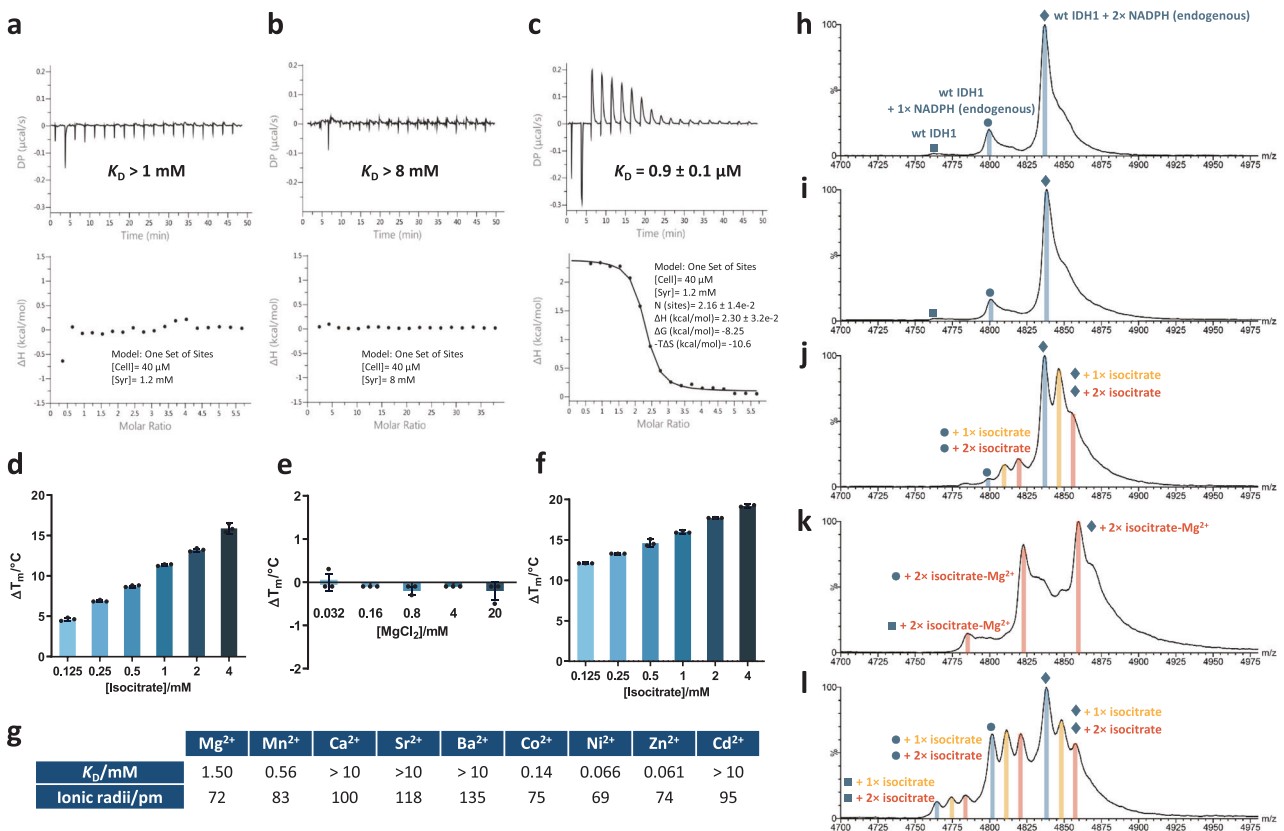

**Fig. 2 Isocitrate binding to wt IDH1 is promoted by Mg²⁺ ions.** ITC analysis of **a** DL-isocitrate, **b** MgCl₂ and **c** DL-isocitrate–Mg²⁺ complex binding to EDTA-treated wt IDH1, in 50 mM Tris-HCl, pH 7.5 (+5 mM MgCl₂ for **c**). Binding of isocitrate or MgCl₂ alone to wt IDH1 is weak ($K_D$s >1 and >8 mM, respectively). Isocitrate–Mg²⁺ complex binding to wt IDH1 is strong ($K_D = 0.9$ μM). Stoichiometry of DL-isocitrate binding to wt IDH1, $n = 2$ as presumably only D-isocitrate binds to wt IDH1[12]. DSF results for **d** DL-isocitrate, **e** MgCl₂ and **f** DL-isocitrate in the presence of 10 mM MgCl₂, with wt IDH1 in 50 mM Tris-HCl, pH 7.5. Thermal stabilisation of wt IDH1 by isocitrate is enhanced by MgCl₂. MgCl₂ alone induces no thermal shift. Data are mean ± SD, $n = 3$ technical replicates. **g** ITC measured dissociation constants ($K_D$) for metal ion binding to DL-isocitrate. Ionic radii of metal ions assume 2⁺ charge state, coordination number 6, and high spin, where applicable[74]. See Supplementary Fig. S9 for ITC plots. **h–l** Non-denaturing MS analysis of EDTA-treated wt IDH1 binding to D-isocitrate, ±Mg²⁺. Data for the IDH1 dimer $m/z = 20^+$ charge state are shown. Cone voltage: 100 V, unless stated. Buffer: 200 mM ammonium acetate, pH 7.5. **h** EDTA-treated wt IDH1 (50 μM) dimer copurifies with two NADPHs. **i** IDH1 + MgCl₂ (50:200 μM). Note the peak shifts to a higher mass with a small increment in the singly charged mass of ~9 Da, possibly reflecting non-specific Mg²⁺ binding. **j** wt IDH1 + D-isocitrate (50:200 μM), showing binding of one (+~193 Da, peak marked with yellow-orange line) and (more weakly) two isocitrates (coral red line). **k** wt IDH1 + D-isocitrate + MgCl₂ (50:200:200 μM). Mg²⁺ addition shifts the equilibrium towards increased isocitrate binding with the complex with two isocitrate–Mg²⁺ (+~433 Da) dominating. **l**, as **k**, with a higher cone voltage (200 V) that induces partial in-source dissociation; complexes with no and single isocitrate bound are now observed, likely produced by dissociation of the complex with two isocitrates. See 'Methods' for details.

the abundance of the two isocitrate-bound wt IDH1 complex (Fig. 2k), indicating strengthening of isocitrate binding to wt IDH1 by Mg²⁺. The mass shifts also suggest that isocitrate–Mg²⁺ binds as a complex to wt IDH1. Note that turnover was not observed, because NADPH instead of NADP⁺ is bound to wt IDH1. Half-site reactivity is reported for wt IDH1, where the addition of both isocitrate and Mg²⁺ promotes the release of one NADPH from the two NADPH bound wt IDH1 to activate the enzyme for catalysis[23]. In agreement with this report, the abundance of one NADPH bound wt IDH1 species increases upon the addition of both isocitrate and Mg²⁺ to wt IDH1 (Fig. 2k), but not when Mg²⁺ or isocitrate was added individually (Fig. 2i, j).

Both 2OG and its close isosteric surrogate, *N*-oxalylglycine (NOG, which inhibits wt and R132H IDH1[28]), bind weakly to both wt IDH1 (as shown by DSF (Supplementary Fig. S8b) and NMR (Supplementary Fig. S12a)) and R132H IDH1 without metal ions (Supplementary Figs. S12b and S13c, d). The addition of Mg²⁺ improves binding of NOG to R132H IDH1, resulting in an increased abundance of the R132H IDH1 dimer with two

NOG–Mg²⁺ complexed (Supplementary Fig. S13e), albeit to a lesser extent compared to isocitrate–Mg²⁺ binding to wt IDH1 (Fig. 2k); indeed, a peak corresponding to R132H IDH1 with no NOG–Mg²⁺ bound is still observed. The combination of 2OG and Mg²⁺ was avoided in binding studies, as it enables catalysis with the copurifying NADPH in R132H (and wt) IDH1 (Supplementary Fig. S14).

The binding of 2HG to wt or R132H IDH1 was the weakest of the tested (potential) substrates; 2HG was not observed to bind to either enzyme without metal ions by DSF (Supplementary Fig. S8b), ITC or NMR (Supplementary Fig. S15). Mg²⁺, Ca²⁺ and Cd²⁺ promote 2HG binding to wt IDH1, although Cd²⁺ but not Mg²⁺ or Ca²⁺ promotes 2HG binding to R132H IDH1 (Supplementary Fig. S15).

**Binding of potent allosteric inhibitors hinders substrate binding to wt IDH1.** We then carried out studies with two potent R132H IDH1 inhibitors, AG-120 (Ivosidenib; IC₅₀ 12 nM[36]) and GSK864 (IC₅₀ 4.6 nM[17]), choosing to focus studies on them

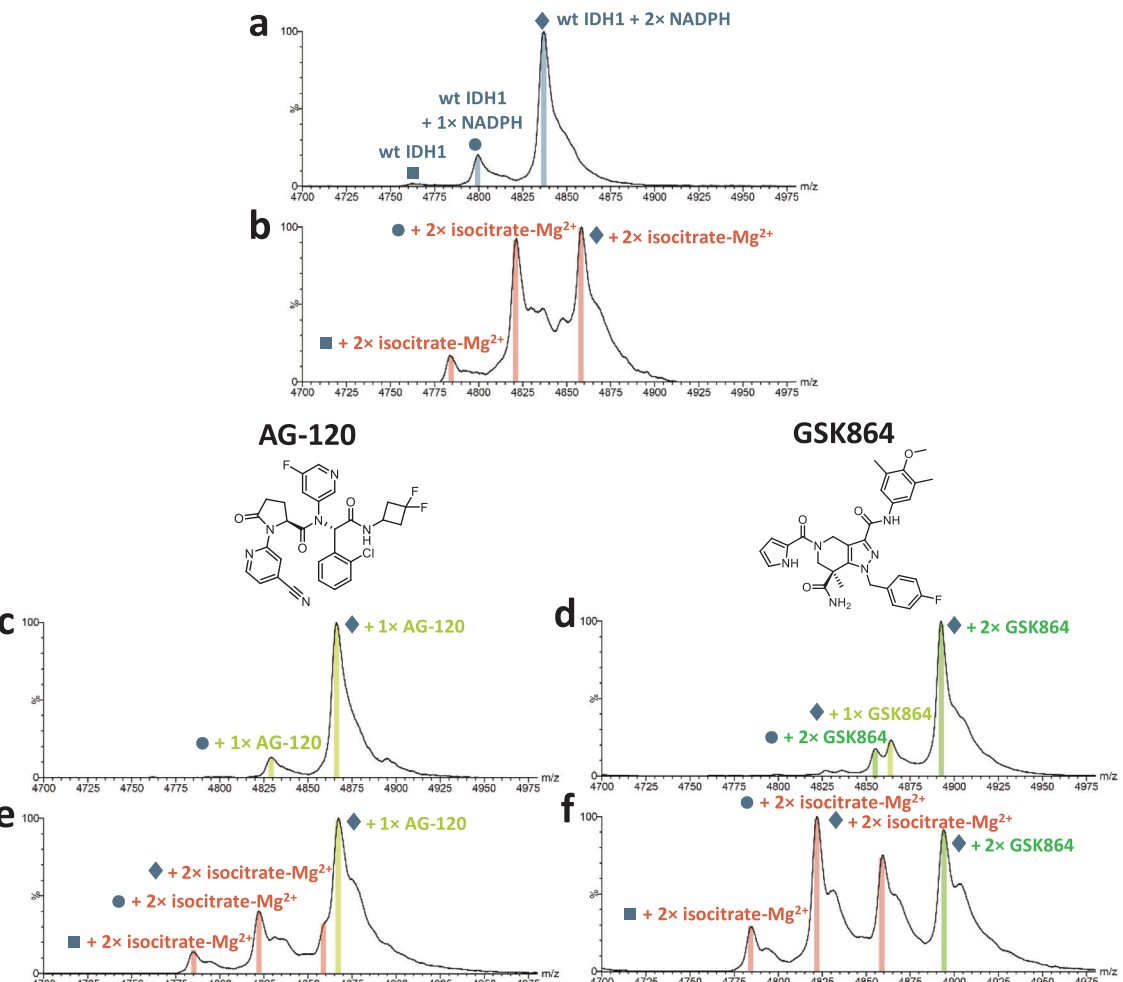

**Fig. 3 Isocitrate–Mg²⁺ binding to wt IDH1 is not fully diminished by R132H IDH1 inhibitors.** Non-denaturing MS analysis of EDTA-treated wt IDH1 with inhibitors. Data for the IDH1 dimer $m/z = 20^+$ charge state are shown. Cone voltage: 100 V. Buffer: 200 mM ammonium acetate, pH 7.5. See 'Methods' for details. **a** wt IDH1 (50 μM) with predominantly two NADPHs bound. **b** wt IDH1 + D-isocitrate + Mg²⁺ (50:200:200 μM) showing two isocitrate–Mg²⁺ bound to wt IDH1 dimer. **c** wt IDH1 + AG-120 (50:200 μM) showing AG-120 bound to wt IDH1 with a predominant stoichiometry of one inhibitor/IDH1 dimer. Note that excess AG-120 does not promote the binding of a second AG-120. **d** wt IDH1 + GSK864 (50:400 μM) showing GSK864 bound to wt IDH1 with a predominant stoichiometry of two inhibitors/IDH1 dimer. The GSK864 concentration (400 μM) was chosen to saturate its binding to wt IDH1 without inducing precipitation. **e** wt IDH1 + D-isocitrate + Mg²⁺ + AG-120 (50:200:200:200 μM) reflecting apparent superposition of **b** and **c**. **f** wt IDH1 + D-isocitrate + Mg²⁺ + GSK864 (50:200:200:400 μM) reflecting apparent superposition of **b** and **d**. In **e**, **f**, wt IDH1 complexed with both isocitrate–Mg²⁺ and R132H IDH1 inhibitor is not observed, indicating wt IDH1 binds to either the isocitrate–Mg²⁺ complex or inhibitor, but not both simultaneously. Note, Mg²⁺ addition does not affect inhibitor binding (Supplementary Fig. S16).

because they have different scaffolds and binding stoichiometries (Fig. 3). Non-denaturing MS reveals that AG-120 binds to wt and R132H IDH1 with a stoichiometry of one inhibitor per IDH1 dimer (with two NADPHs bound) (Figs. 3c and 4b). By contrast, GSK864 binds to wt and R132H IDH1 with a stoichiometry of two inhibitors per IDH1 dimer under our conditions (Figs. 3d and 4c); this stoichiometry is consistent with the reported X-ray crystal structure of its close analogue, GSK321[17].

When AG-120 or GSK864 is added to the D-isocitrate–Mg²⁺–wt IDH1 complex, we observed the formation of a mixture of wt IDH1 bound with either of the inhibitors or two D-isocitrate–Mg²⁺ complexes (Fig. 3e, f). No peak(s) corresponding to wt IDH1 with both isocitrate and inhibitor bound was observed. X-ray[17] and cryo-electron microscopy[19] structures of analogues of GSK864 and AG-120 with R132H IDH1 show that they bind at the dimer interface, whilst isocitrate binds at the active site (Fig. 1b). Although the binding sites do not spatially overlap, the inhibitor and isocitrate were not observed to simultaneously bind to wt IDH1, implying that once the inhibitor

binds at the dimer interface, it disfavours the active site binding of isocitrate. Notably, despite using high inhibitor concentrations (200/400 μM), the inhibitors did not displace all of the isocitrate from wt IDH1 (Fig. 3e, f), reflecting the strong affinity of the isocitrate–Mg²⁺ complex for wt IDH1. In the absence of D-isocitrate, no differences were observed in the MS spectra for inhibitor binding to wt or R132H IDH1 with or without Mg²⁺ (Supplementary Fig. S16), consistent with structural work showing no direct contact between the inhibitors and the active site metal ion (Fig. 1b). Isocitrate–Mg²⁺ binding to wt IDH1 promotes the release of one NADPH from the as-purified two NADPH bound wt IDH1, likely to form the closed active state of the enzyme; by contrast, both inhibitors were observed to selectively bind to the two NADPH-bound forms of wt IDH1 (Fig. 3e, f) and R132H IDH1 (see below), which for wt IDH1 is reported to be an open inactive state[23].

**Inhibitors displace metal ions from R132H IDH1.** Competition experiments involving substrate and inhibitors on R132H IDH1

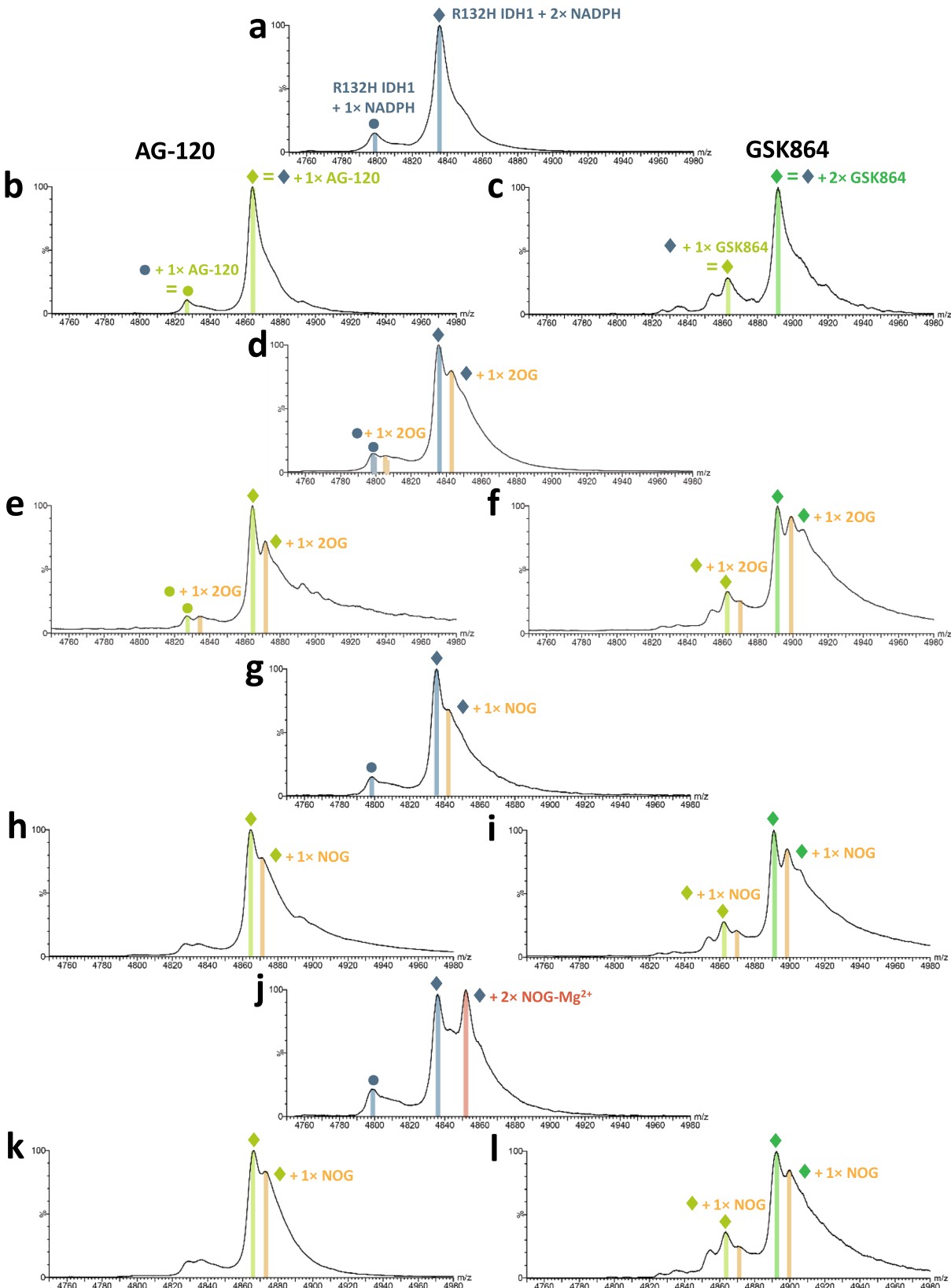

were then conducted using non-denaturing MS. Without divalent metal ions, only one 2OG or NOG molecule was observed to bind to the R132H IDH1 dimer when using a tenfold molar excess relative to the protein (Fig. 4d, g); however, these studies may not fully reflect the protein-bound stoichiometry due to the relatively weak binding of 2OG/NOG to R132H IDH1. The addition of AG-120 or GSK864 does not affect 2OG/NOG binding, consistent with the binding of 2OG and the inhibitors at non-overlapping sites (Fig. 4e, f, h, i). In the presence of MgCl₂, two NOG molecules were observed to bind to each R132H IDH1 dimer, supporting the proposal that Mg²⁺ ions enhance NOG (and by implication 2OG) binding (Fig. 4j). In contrast to the

**Fig. 4 N-Oxalyglycine-Mg$^{2+}$ binding to R132H IDH1 is diminished by R132H IDH1 inhibitors.** Non-denaturing MS analysis of EDTA-treated R132H IDH1 with inhibitors. Data for the IDH1 dimer $m/z = 20^+$ charge state are shown. Cone voltage: 100 V. Buffer: 200 mM ammonium acetate, pH 7.5. See 'Methods' for details. **a** R132H IDH1 (50 μM) with two NADPHs bound. **b** R132H IDH1 + AG-120 (50:200 μM) showing predominantly one AG-120 bound/R132H IDH1 dimer. **c** R132H IDH1 + GSK864 (50:400 μM) showing predominantly two GSK864-bound/R132H IDH1 dimers. **d** R132H IDH1 + 2OG (50:500 μM) showing one 2OG binding weakly to the R132H IDH1 dimer. **e** R132H IDH1 + 2OG + AG-120 (50:500:200 μM), reflecting a shift of protein peak **d** by the AG-120 mass without affecting 2OG binding (without metal ions). **f** R132H IDH1 + 2OG + GSK864 (50:500:400 μM). The binding of GSK864 to R132H IDH1 does not affect 2OG binding (without metal ions). **g** R132H IDH1 + NOG (50:500 μM) showing one NOG binding weakly to R132H IDH1 as indicated by a shoulder peak. **h** R132H IDH1 + NOG + AG-120 (50:500:200 μM), reflecting a shift of **g** by the AG-120 mass without affecting NOG binding, without metal ions (similar to the 2OG case **e**). **i** R132H IDH1 + NOG + GSK864 (50:500:400 μM). Similarly to **h**, the binding of GSK864 does not affect NOG binding, without metal ions. **j** R132H IDH1 + NOG + MgCl$_2$ (50:500:500 μM), showing two NOG–Mg$^{2+}$-bound/R132H IDH1. Mg$^{2+}$ addition significantly enhances NOG binding. **k** R132H IDH1 + NOG + MgCl$_2$ + AG-120 (50:500:500:200 μM). Instead of shifting by the inhibitor mass without affecting the binding of two NOG, the spectrum resembles **h** with only one NOG bound. **l** R132H IDH1 + NOG + MgCl$_2$ + GSK864 (50:500:500:400 μM). Similarly to **k**, the two NOG–Mg$^{2+}$-bound peak is depleted, with only one NOG bound to R132H IDH1. The resemblance of **k** or **l** to **h** or **i** without metal ions suggests that inhibitor binding weakens that of Mg$^{2+}$.

results without Mg$^{2+}$, the addition of the inhibitors to the NOG–Mg$^{2+}$–R132H IDH1 complex resulted in inhibitor-bound R132H IDH1 with the loss of the peaks for the 2× NOG–Mg$^{2+}$ complexes (Fig. 4k, l). Indeed, the spectra of NOG–Mg$^{2+}$–inhibitor mixtures (Fig. 4k, l) resemble those without Mg$^{2+}$ (Fig. 4h, i), inferring that binding of the inhibitors promotes the release of Mg$^{2+}$, which is required for NOG (and by implication productive 2OG) binding.

**NMR studies imply metal ions help isocitrate, but not 2OG, to remain complexed to IDH1 on inhibitor binding.** The effects of AG-120 and GSK864 on substrate binding, with and without metal ions, were then investigated by ligand-observed $^1$H NMR (700 MHz) (Fig. 5). Without metal ions, the D-isocitrate (50 μM) signals are line-broadened by the addition of wt IDH1 (6.25 μM), implying protein binding[37]. Inhibitor addition (50 μM) results in incomplete recovery of the D-isocitrate signals, indicating that release of some, but not all, of the bound D-isocitrate is promoted by binding of the inhibitors (Fig. 5a, b). In the presence of Mg$^{2+}$ or Ca$^{2+}$ (10 mM), however, no isocitrate was observed to be displaced from wt IDH1 upon inhibitor addition (under the tested conditions), consistent with our previous studies showing metal ions are important in stabilising the binding of isocitrate to wt IDH1 (Fig. 2), including in the presence of inhibitors (Fig. 5c–f).

Because 2OG is a weak binder to R132H IDH1, investigations on its binding were difficult without metal ions (Supplementary Fig. S12b). The addition of Ca$^{2+}$ (10 mM) enhances the binding of 2OG (50 μM) to R132H IDH1 (50 μM); as before, the combination of 2OG and Mg$^{2+}$ was avoided in binding studies as it enables catalysis. Inhibitor addition (50 μM), however, displaces most of the 2OG in the presence of Ca$^{2+}$ (Fig. 5g, h). Experiments with NOG and Mg$^{2+}$ gave analogous results (Fig. 5i, j).

Consistent with the MS studies (Figs. 3 and 4), the NMR results suggest that with Mg$^{2+}$ or Ca$^{2+}$, the presence of AG-120 or GSK864 weakens the binding of 2OG/NOG with R132H IDH1, but not of D-isocitrate from wt IDH1 (or at least less strongly). Removing the Mg$^{2+}$ or Ca$^{2+}$ ions enables displacement of D-isocitrate from wt IDH1 by the inhibitors. Their different interactions with metal ions thus (in part) account for the wt/mutant selectivity of IDH1 inhibitors, i.e. they are selective for R132H as they can displace 2OG/Mg$^{2+}$ from R132H IDH1 more easily than displacing the isocitrate–Mg$^{2+}$ complex from wt IDH1.

**Metal ion concentration affects the potency of R132H IDH1 inhibitors.** The combined non-denaturing MS and NMR results imply that the inhibitors likely work by binding to an inactive

form of the enzyme and by displacing Mg$^{2+}$, which is important for productive substrate binding. It follows that increasing the Mg$^{2+}$ concentration may make R132H IDH1 more resilient to inhibition. Hence, we conducted inhibition studies to investigate the extent of inhibition of R132H IDH1 by GSK864, AG-120 and its less potent analogue AGI-5198 (IC$_{50}$ 70 nM[38]), using various MgCl$_2$ concentrations. The results reveal that the degree of inhibition decreases with increasing MgCl$_2$ concentration (Fig. 6a), particularly over the range of 2.5–20 mM, which includes the MgCl$_2$ $K_M$ (4.4 mM) for R132H IDH1, supporting the proposal that inhibitor binding weakens Mg$^{2+}$ binding. In the presence of a physiologically relevant concentration (150 mM) of NaCl, the IC$_{50}$ values of AGI-5198 for R132H IDH1 increase as the MgCl$_2$ concentrations increase from 1.25 to 80 mM (Fig. 6b, c).

Several studies suggest that variations in intracellular concentrations of Mg$^{2+}$ are linked to healthy physiology (e.g. circadian rhythm) and diseases[39–41]. We thus investigated whether different Mg$^{2+}$ concentrations may affect R132H IDH1 inhibition in cells, by studies with LN18 glioblastoma cells[42] bearing R132H IDH1 and manifesting high 2HG levels. The MgSO$_4$ concentration of the cell culture medium (originally 0.8 mM) was reduced over 3 weeks to 0.05 mM, to enable cell viability at low Mg$^{2+}$ levels. The LN18 cells were then treated with a dimethyl sulfoxide (DMSO) control, AGI-5198 or GSK864 in Dulbecco's modified Eagle's medium (DMEM) containing 0.05–10 mM MgSO$_4$ for 19 h. Filtered cell extracts were analysed for their 2HG abundance by anion-exchange chromatography mass spectrometry (AEC-MS)[43] and/or Mg$^{2+}$ content by cation-exchange chromatography with conductivity detection (CEC-CD). Consistent with reported results[17,44], both inhibitors substantially suppressed 2HG production by R132H IDH1 at all the tested concentrations of added MgSO$_4$ (Supplementary Fig. S17a, b). No clear correlation between MgSO$_4$ concentration and the degree of R132H IDH1 inhibition (as determined by changes in 2HG levels with inhibitor treatment relative to the DMSO control), however, was observed, possibly due to Mg$^{2+}$ homeostasis mechanisms[45]. Indeed, intracellular Mg$^{2+}$ was observed at similar levels across all samples despite large extracellular changes in the MgSO$_4$ concentrations (0.1, 2.5 and 10 mM) (Supplementary Fig. S17c).

## Discussion

Near all potent R132H IDH1 inhibitors with reported structures are allosteric in nature, binding at the dimer interface (Fig. 1b). Prior studies have not explained this observation, nor why the inhibitors are selective for R132H IDH1. A study concerning GSK321 (IC$_{50}$ 4.6 nM for R132H IDH1) reported it to be competitive with respect to the substrate 2OG, and noncompetitive

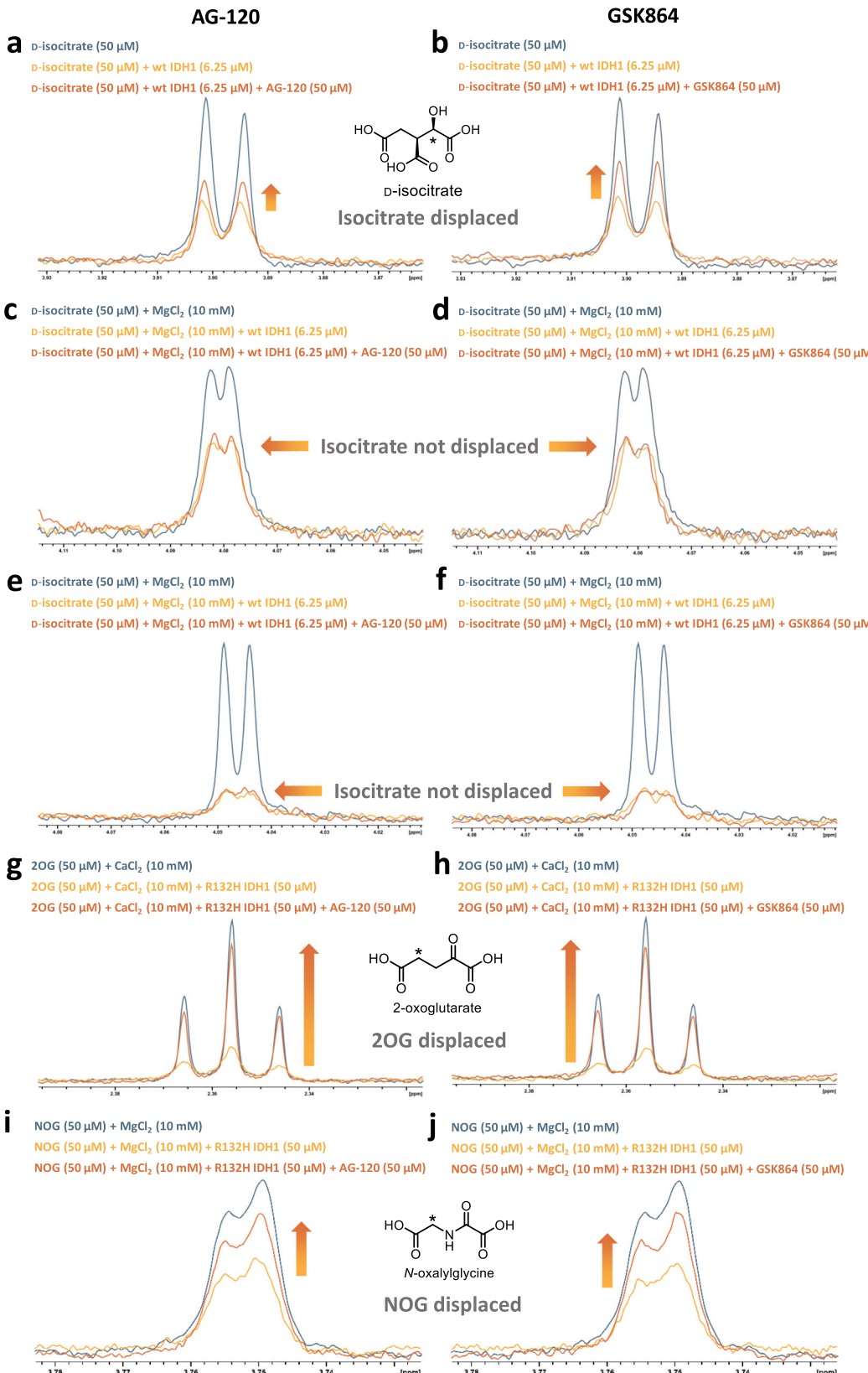

with respect to NADPH[17]. ML309 and AGI-5198, inhibitors with a different scaffold from GSK321, were also reported to be competitive with 2OG[26]. Prior studies[21,27] and our investigations are consistent in finding >10$^2$-fold lower apparent $K_M$ values for both Mg$^{2+}$ and Mn$^{2+}$ for wt IDH1 compared to R132H IDH1. Although, in some cases, disrupted Mg$^{2+}$ binding has been considered in the mechanisms of the allosteric inhibitors[27], to our knowledge the role of this in determining selectivity has not been explored. Previous studies on human IDH1 have not taken into account the pioneering observations of Colman and co-workers providing evidence that the preferred substrate for native porcine heart IDH2 is the D-isocitrate–Mg$^{2+}$ complex[25].

**Fig. 5 R132H IDH1 inhibitors do not efficiently displace isocitrate from wt IDH1 in the presence of metal ions, but displace 2OG from R132H IDH1.** CPMG-edited $^1$H NMR binding analyses of D-isocitrate binding to wt IDH1, or 2OG binding to R132H IDH1, in the presence of inhibitors. Buffer: 50 mM Tris-D$_{11}$-HCl, pH 7.5 in 90% H$_2$O/10% D$_2$O (v/v). See 'Methods' for details. **a** D-isocitrate binding to EDTA-treated wt IDH1 (D-isocitrate:wt IDH1, 8:1) in the presence of AG-120, without metal ions. **b**, as **a**, but with GSK864. D-Isocitrate is partially displaced from wt IDH1 by AG-120 and GSK864 without metal ions. **c**, **d**, as **a**, **b**, but with 10 mM MgCl$_2$. **e**, **f**, as **a**, **b**, but with 10 mM CaCl$_2$. D-isocitrate is displaced from wt IDH1 by the inhibitors without metal ions, but not in the presence of MgCl$_2$ or CaCl$_2$. **g** 2OG binding to EDTA-treated R132H IDH1 (2OG:R132H IDH1, 1:1) in the presence of 10 mM CaCl$_2$ and AG-120. **h**, as **g**, but with GSK864. **i**, **j**, as **g**, **h**, but replacing 2OG and CaCl$_2$ with NOG and MgCl$_2$. 2OG or its surrogate NOG is displaced from R132H IDH1 by the inhibitors in the presence of metal ions. Blue: substrate or substrate–metal complex. Yellow-orange: substrate or substrate–metal complex with IDH1. Orange: substrate or substrate–metal complex with IDH1 and inhibitor.

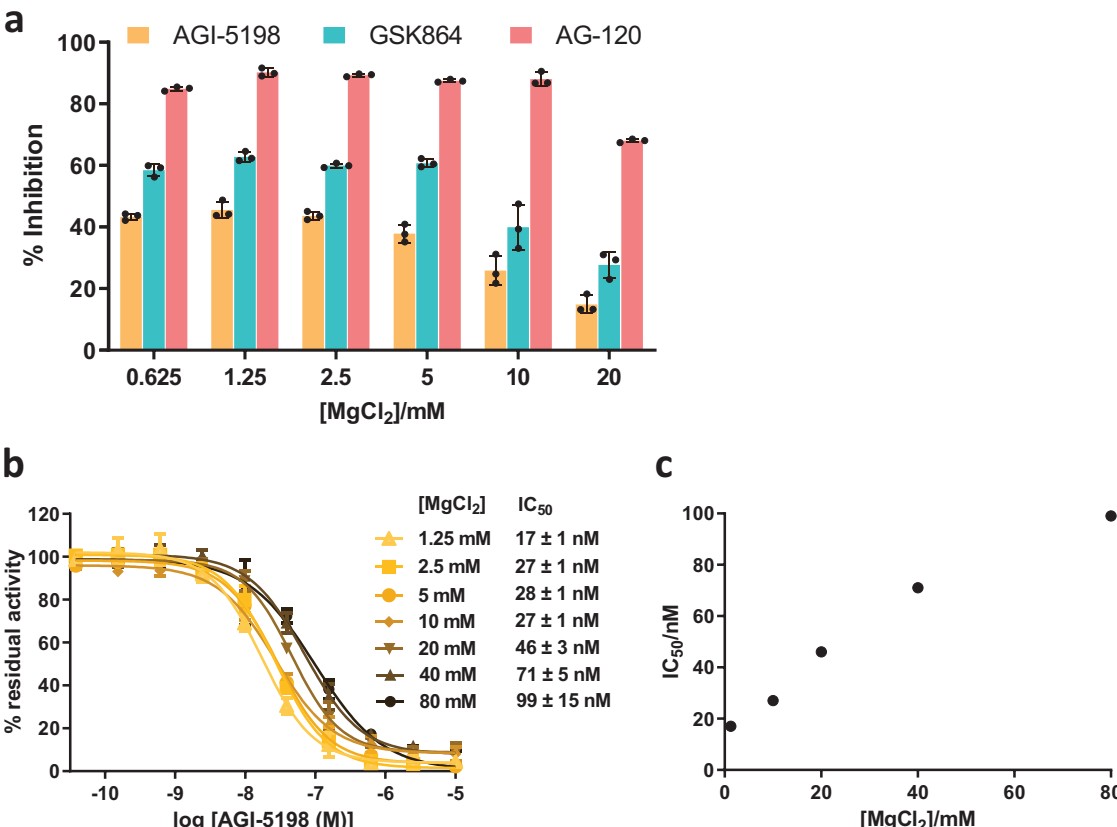

**Fig. 6 Studies on R132H IDH1 inhibition with an isolated enzyme in the presence of different Mg$^{2+}$ concentrations. a** The effects of different MgCl$_2$ concentrations on the extent of inhibition of R132H IDH1 (300 nM) by its inhibitors (100 nM), AG-120, GSK864 (different scaffold and binding stoichiometry from AG-120) and AGI-5198 (which has similar scaffold to AG-120, but which is less potent[75]), as determined by absorbance assays. **b**, **c** IC$_{50}$ of AGI-5198 for R132H IDH1 at different MgCl$_2$ concentrations in the presence of 150 mM NaCl. IC$_{50}$ values of AGI-5198 for R132H IDH1 increase with increasing MgCl$_2$ concentrations, as determined by absorbance assays. Data are mean ± SD, $n = 3$ technical replicates. See 'Methods' for assay conditions.

Our ITC analyses show that Mg$^{2+}$ is essential for D-isocitrate binding to human wt IDH1, as we did not observe binding of isocitrate or Mg$^{2+}$ in isolation to wt IDH1 (Fig. 2a–c). Analogous results were obtained for Mn$^{2+}$, although in the cellular context, Mg$^{2+}$ is likely more relevant. Substantial enhancement of D-isocitrate binding to wt IDH1 and of 2OG/NOG to R132H IDH1 by metal ions was validated by complementary methods including DSF, non-denaturing MS and NMR (Fig. 2d–f, h–l and Supplementary Figs. S12 and S13). An important difference between wt and R132H IDH1, supported by studies in solution and on enzyme complexes, is that the isocitrate–Mg$^{2+}$ complex is the preferred wt IDH1 substrate. However, for R132H IDH1, since 2OG and Mg$^{2+}$ only interact weakly in solution, the evidence implies 2OG and Mg$^{2+}$ bind separately (at least predominantly) to R132H and less strongly than the isocitrate–Mg$^{2+}$ complex to wt IDH1.

As described below these observations have implications for the mechanism by which selective inhibition for R132H over wt IDH1 is achieved for AG-120, GSK864 and, by implication, related allosteric inhibitors. The results are also of interest with respect to fundamental aspects of cell metabolism. Our conclusion that the D-isocitrate–Mg$^{2+}$, or less likely the D-isocitrate-Mn$^{2+}$, complex is the preferred human wt IDH1 substrate is the same as that reached previously for native porcine heart IDH2[25], suggesting that it may be a conserved feature of IDH orthologues. This and the precise nature of the D-isocitrate divalent metal ion and related complexes (e.g. with citrate) in solution are under current investigation—given its potential importance for much of biology the interactions between metal ions and di-/tricarboxylic metabolites appear to be an understudied field[46,47]. Approximately 70% of enzymes are predicted to bind metal ions for structural stabilisation and/or catalytic function[48]; although in some cases the order of metal ion binding during catalysis is established, in many cases including kinases and nucleases where the substrates also bind metal ions, it is often unclear[49–51]. The formation of small-molecule carboxylic acid–metal ion complexes

is of potential relevance to the origin of metabolism—a recent study has shown how $Fe^{2+}$ can drive the redox reaction-mediated formation of multiple biologically relevant metabolites, including TCAs from pyruvate or glyoxylate[52]. It is possible that, in addition to redox-active complexes, the formation of non-redox-active complexes with metal ions regulates the kinetics of these transformations, so potentially helping to enable the construction of robust reaction cycles.

Our analyses involving $Mg^{2+}/Mn^{2+}$ provide a mechanistic basis for selectivity of the R132H IDH1 inhibitors over wt IDH1, both in isolated form and in cells. We chose two representative R132H IDH1 inhibitors with different scaffolds and binding stoichiometries for study, i.e. AG-120, which is structurally analogous to AGI-5198 and ML309, and GSK864, which is a close analogue of GSK321[17,36]. Non-denaturing MS studies showed that in the presence of $Mg^{2+}$, binding of isocitrate and inhibitor to wt IDH1 is mutually exclusive and isocitrate binding is not substantially reduced by the presence of the inhibitor (Fig. 3). NMR studies also demonstrate that isocitrate is not displaced by inhibitor binding in the presence of $Mg^{2+}/Ca^{2+}$ (Fig. 5), but can be partially displaced in the absence of metal ions. Thus, the R132H IDH1 selective inhibitors are weakly active against wt IDH1 (at least in substantial part) because they cannot easily displace isocitrate–$Mg^{2+}$ from the isocitrate–$Mg^{2+}$–wt IDH1 complex (Fig. 7).

The concentration of free cytosolic $Mg^{2+}$, which represents only a small fraction (1–5%) of the total $Mg^{2+}$ concentration in cells, is estimated to be 0.22–1 mM[40,53,54], and in particular, 0.3 mM for the human brain[41], although this may vary locally and in diseased states as two studies showed elevated $Mg^{2+}$ in brain tumours compared to normal tissue[41,55]. The intracellular concentration of isocitrate is reported to be 29 μM in rat liver, 34 μM in rat heart[56] and 0.57 mM in *Escherichia coli* cells[57]. The available evidence thus implies that for wt IDH1, the $K_M$ of $Mg^{2+}$

(36 μM, Table 1) and the $K_D$ of isocitrate (0.9 μM, Fig. 2c) in the presence of $Mg^{2+}$ are both substantially lower than cellular levels of free $Mg^{2+}$ and isocitrate. Consequently, the isocitrate–$Mg^{2+}$ complex should be readily available for wt IDH1 catalysis in cells (although, of course, its concentration may vary within cells).

The results imply that selectivity for R132H IDH1 over wt IDH1 substantially arises because of the differential effects of the inhibitors on metal ion and substrate binding by R132H compared to wt IDH1, despite the inhibitors binding to a likely inactive form of both enzymes and the inhibitor binding site being relatively remote from the R132H substitution. Although perturbed metal ion binding has been previously considered in terms of R132H inhibition[27], its relationship to substrate binding by wt and R132H IDH1 has not been explored. The observations imply that the preferred substrate for wt IDH1 is the relatively tight-binding isocitrate–$Mg^{2+}$ complex ($K_D = 0.9$ μM, Fig. 2c), whereas for R132H IDH1, the preferred mechanism involves separate and weaker binding of 2OG and $Mg^{2+}$ (Figs. 4 and 5), so rationalising why the inhibitors are selective for R132H over wt IDH1. The apparent $K_M$ of $Mg^{2+}$ (4.4 mM, Table 1) for R132H IDH1 is higher than the free cytosolic $Mg^{2+}$ concentration and 2OG does not tightly bind $Mg^{2+}$ (Supplementary Fig. S10). Note that we cannot entirely rule out the binding of the 2OG–$Mg^{2+}$ complex to R132H IDH1, although we have no direct evidence for this. The differences in substrate and $Mg^{2+}$ interactions with wt and R132H IDH1 thus enable selective R132H inhibition both when using isolated enzymes and in cells (Fig. 7).

The question then arises as to the structural basis of the kinetic observations—this is of interest given that the R132H substitution and substrate are at the active site and the inhibitors bind at the dimer interface (Fig. 1b). Investigations on the mechanism of R132H IDH1 inhibitors using non-denaturing MS show that, in all cases, the addition of inhibitors to a mixture of the substrate,

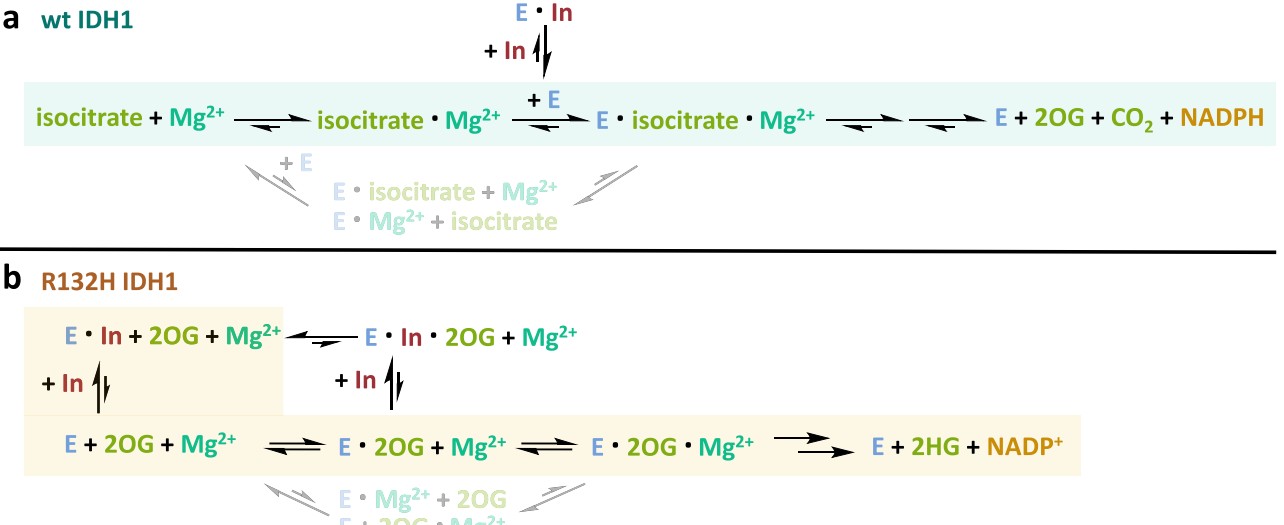

**Fig. 7 Summary of kinetic proposals for the formation of catalytically productive wt and R132H IDH1 metal ion–substrate complexes and for the selective allosteric inhibition of R132H IDH1. a** Catalysis by wt IDH1. The preferred reaction pathway involving binding of the isocitrate–$Mg^{2+}$ complex to E is shaded. E = wt IDH1, which ultimately must be loaded with one $NADP^+$ to satisfy the observed half-site reactivity[23]. Note the precise timings/mechanisms of NADP(H) binding/release during catalysis are not fully established, although the inhibitors can bind to the dimeric form of wt IDH1 with two NADPHs bound. Isocitrate–$Mg^{2+}$ binding to wt IDH1 promotes the release of one NADPH from the as-purified two NADPH bound wt IDH1 to form a closed active state[23]. **b** Catalysis and inhibition of R132H IDH1. The likely preferred pathways for a reaction involving separate binding of 2OG and $Mg^{2+}$ (since 2OG and $Mg^{2+}$ interact only weakly) and for inhibition of R132H IDH1 are shaded. The allosteric inhibitors bind preferentially to the two NADPH-bound forms of R132H IDH1, which is an open inactive state[23]; they thus bias IDH1 into an inactive conformation. Note, the binding of $Mg^{2+}$ and 2OG to R132H IDH1 is weaker than that of the isocitrate–$Mg^{2+}$ complex to wt IDH1. E = R132H IDH1 loaded with (at least) one NADPH. In = allosteric IDH1 inhibitor.

metal ion and enzyme does not displace the copurifying NADPH in wt or R132H IDH1, consistent with the conclusion from kinetic studies[17,26] that the inhibitors are not competitive with respect to NADP(H). The MS studies demonstrate that in the absence of metal ions, the degree of binding of substrate, 2OG or its surrogate NOG is apparently (at least substantially) unaffected by inhibitor binding to R132H IDH1. This finding may appear to conflict with reported kinetic studies[17,26], which conclude that some R132H IDH1 inhibitors are competitive with 2OG. However, the kinetic studies are necessarily done in the presence of catalytically active $Mg^{2+}/Mn^{2+}$; binding studies can be conducted with or without metal ions to dissect the roles of $Mg^{2+}$. In the presence of $Mg^{2+}$, R132H substrate–$Mg^{2+}$ binding is weakened with the loss of $Mg^{2+}$ ion on inhibitor addition. Overall, our studies provide evidence to support the proposal that the inhibitors do not directly compete with the substrate, but are competitive with respect to $Mg^{2+}$, which is required for productive 2OG binding to R132H IDH1.

Crystal structures of inhibitors complexed with R132H IDH1[13,17,18,27,58,59] (Fig. 1b) show that the inhibitors and $Mg^{2+}$ do not directly interact. Instead, they bind on opposite sides of α-helix 10 (residues N271–G286), from which D275 and D279 act as metal ion-chelating ligands (along with D252 from α-helix 9 of the other monomer subunit)[13] (Supplementary Fig. S18a). The dimer interface is formed predominantly by four laterally arranged α-helices (α9 and α10 from each monomer of the homodimer) (Supplementary Fig. S18b). All available structures for allosteric R132H IDH1 inhibitors reveal they form one or more hydrogen bonds with residues from α10, e.g. Q277 (Vorasidenib)[58], S278 (Novartis 889)[59], D279 (Sanofi 1)[27], S280 (BAY-1436032 analogue)[18] and V281 (to its backbone by GSK321)[17] (Supplementary Fig. S18a). Although the precise details especially of the dynamics involving the inhibitor–α10–$Mg^{2+}$-substrate interaction cascade (which may relate to motions remote from the active site) remain to be defined, the combined results imply interaction of the inhibitors with α10 results in the displacement of the residues involved in $Mg^{2+}$ coordination by R132H IDH1, which in turn disrupts productive 2OG binding at the active site. Weakening of substrate and $Mg^{2+}$ binding promotes the formation of the open inactive conformation of (R132H) IDH1[12] complexed with two NADPH molecules, as observed in crystal structures of inhibitor-bound R132H IDH1[17,27,58]. It is of interest that in none of the reported structures of allosteric inhibitor-bound R132H IDH1, has a divalent metal ion been observed at the active site, despite the presence of high concentrations (~100 mM) of $Ca^{2+}$ (BAY-1436032 analogue)[18] or $Mg^{2+}$ (Vorasidenib)[58] in the crystallisation solution. This observation is consistent with our proposal that allosteric inhibitor binding to R132H IDH1 weakens the binding of $Mg^{2+}/Mn^{2+}$ ions.

The knowledge that the existing R132H IDH1 inhibitors work by a subtle allosteric mechanism involving disrupting $Mg^{2+}$, and in turn, substrate binding, which differs in detail for wt IDH1, has implications for the development of improved IDH variant inhibitors. The variety of IDH variant inhibitor scaffolds discovered to date[15] may reflect the dynamic nature of the dimer interface, which is linked to active site metal ion and substrate binding and which may be involved in other correlated motions during catalysis, possibly relating to the half-site reactivity of IDH1[23]. Whilst the dimer interface inhibitor binding mode is clearly efficacious in vivo and, at least in some cases, enables selectivity versus wt IDH1, its mobile nature may make it more prone to resistance (as has been observed[22]) than more typical active site binding/substrate competitive inhibitors. Thus, we suggest future screens for IDH1 variant inhibitors, including aiming to identify inhibitors that work by direct competition with $Mg^{2+}$ and/or 2OG (which might be less prone to resistance)

could employ the use of different $Mg^{2+}$ concentrations and binding, as well as turnover-based assays. Importantly, it should be noted that the allosteric nature of binding of the current inhibitors means competition with 2OG, 2HG and isocitrate, which collectively are present at a high intracellular concentration, can be avoided. Likely more ambitiously than inhibition, the development of small molecules that enhance binding of isocitrate/$Mg^{2+}$ to R132H relative to 2OG/$Mg^{2+}$ to R132H might be explored with a view to restoring wt function to IDH variants (ideally whilst simultaneously inhibiting 2OG reduction). Allosteric-type binding (i.e. not overlapping with that of the substrate) of potential enhancers will be required in this objective. In relation to this, the potential of uncompetitive inhibitors (i.e. those binding to an enzyme–substrate complex) to distinguish between the same enzyme in the presence of different substrates has long been recognised[60], but seldom exploited in drugs. The ability of modern methods for analysing small-molecule binding to proteins, notably MS as employed in our IDH1 work has substantial potential for efficiently identifying allosteric/uncompetitive inhibitors and enhancers, as also shown in prior work concerning protease inhibition[61]. The results also suggest the possibility of natural in vivo regulation of IDH activity by allosteric regulation, as precedented for other oligomeric metabolic enzymes[62].

There is evidence that variations in $Mg^{2+}$ concentrations correlate with physiologically relevant processes (e.g. circadian rhythm[39]) and differences in $Mg^{2+}$ levels are reported in diseases including diabetes[40,41,63]. The observations that wt IDH1 has a lower $K_M$ for $Mg^{2+}$ than R132H IDH1 (Table 1) and that the potency of the R132H inhibitors increases as the $Mg^{2+}$ concentration decreases (Fig. 6) raise the possibility of a therapy for R132H IDH1 bearing cancers in which inhibitor treatment is combined with restriction of $Mg^{2+}$ (possibly by diet therapy or inhibitors blocking $Mg^{2+}$ uptake). We investigated this by studies on the effect of the inhibitors on 2HG levels in R132H bearing cells under conditions of limited $Mg^{2+}$ availability. The results consistently validated the reported ability of the inhibitors to reduce 2HG levels[17,44] (Supplementary Fig. S17a, b); however, no clear evidence for an enhanced decrease in 2HG levels in the cells depleted in $Mg^{2+}$ relative to normal $Mg^{2+}$ conditions was observed, likely due to the operation of efficient $Mg^{2+}$ homoeostasis[64].

The use of altering biological metal ion concentrations for therapy is established; examples include the treatment of iron overload disease (haemochromatosis) with chelators[65] and the historical use of $Co^{2+}$ to treat anaemia[66]. The 2OG-dependent hypoxia-inducible factor (HIF) hydroxylases regulate the levels and activities of HIF; their inhibition upregulates levels of active HIF and its gene targets including that encoding for erythropoietin, in turn increasing red blood cell production[67]. It is proposed that $Co^{2+}$ competes with $Fe^{2+}$ for binding to the active site of the HIF hydroxylases[68], in accord with the therapeutic effect of $Co^{2+}$ [66]. The observation that $Co^{2+}$ inhibits both wt and R132H IDH1 (Supplementary Fig. S5c, d) suggests that (in addition to direct HIF hydroxylase inhibition) depletion of 2OG (or NADPH) levels by $Co^{2+}$ mediated IDH inhibition and hence impaired HIF hydroxylase activity might play a role in $Co^{2+}$-promoted erythropoiesis.

Overall, although our results with cell culture do not demonstrate the feasibility of decreasing extracellular $Mg^{2+}$ concentration to increase R132H IDH1 inhibition, there would seem to be scope for further exploration of metal ion therapy (including potentially via diet therapy and/or circadian control[39]) for cancer treatment, including with respect to IDH variants. Given that there is a very large number of possible mechanisms by which altered metal ion concentrations might affect the regulation of

transcription and other processes, it is suggested that exploration of such therapies should principally be empirically guided.

## Methods

**Recombinant protein preparation.** Recombinant wt and R132H IDH1 homodimers were produced similarly to the reported procedures[12]. In brief, DNA encoding for wt or R132H IDH1 with a His6 tag at their C terminus was inserted into the pET-22b(+) vector. Expression trials using *Escherichia coli* BL21(DE3) pLyS cells, using 2TY medium supplemented with 50 μg mL$^{-1}$ ampicillin and 34 μg mL$^{-1}$ chloramphenicol, were performed at different temperatures (18, 30 and 37 °C) using various isopropyl-β-D-1-thiogalactopyranoside (IPTG) concentrations (0, 0.1, 0.2, 0.5, 0.7 and 1 mM) and different induction times (6, 12 and 20 h). Cells from small-scale production were lysed using the BugBuster kit; preferred conditions for expression were assessed by sodium dodecyl sulfate-polyacrylamide gel electrophoresis (SDS-PAGE) and employed for large-scale expression work. Cells were grown (37 °C) until they reached OD$_{600}$ = 0.6–0.8. Expression was induced by 1 mM IPTG (18 °C, 12 h). After centrifugation (Beckman-Coulter, J-Lite® JLA-10.500, 500 mL centrifuge tubes, 11,876 × *g*/ 8000 r.p.m., 10 min, 6 °C), cells were resuspended in 50 mL of lysis buffer (20 mM Tris-HCl, 500 mM NaCl, pH 7.4) supplemented with DNase I, tris(2-carboxyethyl) phosphine, lysozyme and a tablet of EDTA-free protease inhibitors (Merck). Cells were lysed by sonication on ice. After centrifugation (Beckman-Coulter, JA-25.50, 50 mL tubes, 69,486 × *g*/24,000 r.p.m., 30 min, 6 °C), cell lysates were loaded onto a 5 mL HisTrap HP column (GE Healthcare Life Sciences, Little Chalfont, UK), with 50 mM Tris-HCl, 500 mM NaCl, pH 7.4, containing 20 mM imidazole and then eluted with an imidazole gradient (up to 500 mM imidazole). Fractions containing recombinant IDH1 were purified using a Superdex S200 column (300 mL) equilibrated with 20 mM Tris-HCl, pH 7.4, 100 mM NaCl. Fractions containing the purified protein were concentrated by centrifugal ultrafiltration. The proteins free of divalent metal ions were generated by overnight treatment at 4 °C with EDTA (2000-fold molar excess of protein), followed by purification using a PD-10 desalting column. The purity of the resulting proteins was assessed as >95% (by SDS-PAGE gel, Supplementary Fig. S19) and their concentrations were determined using an ND-1000 NanoDrop spectrophotometer.

**Absorbance assays.** IDH1 activity was measured spectrophotometrically by monitoring NADPH absorbance at 340 nm[12]. Reactions were monitored at 25 °C in 96-well half area clear microtiter plates (Greiner Bio-One 675001) in a final volume of 100 μL continuously over 15 min using a PHERAstar FS Microplate Reader. The standard assay buffer consists of 100 mM Tris-HCl, 10 mM MgCl$_2$, 0.005% Tween-20, 0.1 mg mL$^{-1}$ bovine serum albumin and 0.2 mM dithiothreitol, pH 8.0. The standard reaction mixture contains 4.5 nM wt IDH1, 150 μM isocitrate and 75 μM NADP$^+$ for the oxidative reaction catalysed by wt IDH1 and 400 nM R132H IDH1, 1.5 mM 2OG and 50 μM NADPH for the reductive reaction catalysed by R132H IDH1, unless otherwise stated. All experiments were performed in (at least) triplicate.

Additional conditions for enzyme kinetics were 400 nM R132H IDH1, 1.5 mM isocitrate and 75 μM NADP$^+$ for oxidative reaction by R132H IDH1; 100 nM wt IDH1, 15 mM 2OG, 500 μM NADPH and 100 mM NaHCO$_3$ for reductive reaction by wt IDH1. $K_M$ values were determined while varying concentrations of the substrate/cofactor under investigation while maintaining the concentrations of other components using standard conditions. The reaction rate (μM s$^{-1}$) was calculated from the linear range of the reaction profile as $\Delta A_{340}$/ 6.22 mM$^{-1}$ cm$^{-1}$ (extinction coefficient of NADPH)/0.509 cm (path length) × 1000/time (in s). Graphs were fitted using GraphPad Prism to obtain kinetic parameters.

Inhibition assays of wt or R132H IDH1 with metal ions were conducted with a 12 min pre-incubation of metal ions (25 μL, final concentration 10 mM) and enzyme (25 μL) before the addition of substrate (25 μL) and cofactor (25 μL). The percentage inhibition of R132H IDH1 by commercial inhibitors at different MgCl$_2$ concentrations were measured by diluting the 10 mM inhibitor stock solution in DMSO to 4× final concentration in the standard assay buffer, incubating with R132H IDH1 (25 μL, final concentration 300 nM) for 12 min, followed by the addition of 2OG and NADPH to initiate the reaction. IC$_{50}$ measurements of AGI-5198 for R132H IDH1 at different MgCl$_2$ concentrations were conducted by monitoring reactions in 384-well F-bottom clear plates (Greiner Bio-One 781801) in a final volume of 60 μL continuously over 4 h, with the addition of 150 mM NaCl in the standard assay buffer. AGI-5198 (10 μL, 6× final concentration) and R132H IDH1 (30 μL, final concentration 30 nM) were incubated for 12 min before the addition of a mixture of 2OG and NADPH (20 μL) to initiate the reaction. The difference in absorbance, $\Delta A_{340}$, in the linear range of the reaction profile was converted to % residual activity with the DMSO control being the 100% residual activity reference. The % inhibition is calculated by (1 − activity with inhibitor/ activity with DMSO control) × 100%.

**DSF studies.** Thermal shift experiments were carried out using Sypro Orange dye and a CFX96 Touch™ Real-Time PCR Detection System (Bio-Rad)[69]. A 96-well white PCR plate was used, in which each well contained 20 μL of 2.5 μM wt or R132H IDH1 with 5× Sypro Orange. The standard assay buffer contains 50 mM Tris-HCl at pH 7.5. Various concentrations of substrates were mixed with IDH1

for thermal stabilisation studies. The samples in triplicates were subjected to temperature increases from 20 to 95 °C at 0.2 °C min$^{-1}$. Bio-Rad CFX Manager was used to determine the melting temperature, $T_m$.

**ITC studies.** ITC experiments were conducted using a Malvern MicroCal PEAQ-ITC Automated machine. In general, the protein was prepared in 50 mM Tris-HCl ± 5 mM metal chloride salts at pH 7.5; substrates were dissolved in the same buffer. Four hundred microlitres of cell solution and 200 μL of titrant were transferred to an ITC plate. Titrations were conducted at 25 °C with an initial delay of 60 s and a stir speed of 750 r.p.m. The injection setup consisted of 19 injections that have one initial injection of 0.4 μL and 18 injections of 2.0 μL each with 150 s between the injections. The thermodynamic parameters were obtained using the MicroCal PEAQ-ITC analysis software.

**NMR studies.** NMR studies employed a Bruker AVIII 700 MHz NMR spectrometer with a 5-mm inverse triple-carbon-inverse (TCI) cryoprobe using 3 mm MATCH tubes (Cortecnet). $^1$H Carr-Purcell-Meiboom-Gill (CPMG) NMR experiments (at 298 K) were conducted using the PROJECT-CPMG sequence[70] with τ delays of 0.5 ms and a total filter time (τ × 4 × *N*) of 40 ms. Additional solvent presaturation was applied for 2 s. The assay mixture contains 160 μL of a 50 μM solution of the compound of interest ± 10 mM metal chloride salts in 50 mM Tris-D$_{11}$-HCl in 90% H$_2$O/10% D$_2$O at pH 7.5. Additional protein or inhibitor was added to the mixture with only a small increase (<1.5%) in the total volume. Data were processed with the TopSpin 3.1 software.

Reaction monitoring employed the same spectrometer with a 5-mm inverse TCI cryoprobe using 5 mm tubes (Norell). Data were recorded with a relaxation delay of 2 s and 32 scans, employing a pulse sequence with water suppression ($^1$H excitation sculpting suppression with perfect echo using a 2 ms Sinc selective inversion pulse). There is a time delay of 184 s between acquired spectra over the time-course experiment. Conversion of isocitrate to 2OG by wt IDH1 was monitored using 500 μL of 6 nM wt IDH1, 550 μM (effectively ~400 μM due to water content) DL-isocitrate or 200 μM D-isocitrate, 625 μM NADP$^+$, 10 mM MgCl$_2$, in 50 mM Tris-D$_{11}$-HCl in 90% H$_2$O/10% D$_2$O at pH 7.5. Conversion of 2OG to 2HG by R132H IDH1 was monitored using 500 μL of 1 μM R132H IDH1, 1.5 mM 2OG, 1.5 mM NADPH, 10 mM MgCl$_2$, in 50 mM Tris-D$_{11}$-HCl in 90% H$_2$O/10% D$_2$O at pH 7.5.

**Non-denaturing protein MS.** Non-denaturing mass spectra were obtained using a quadrupole time-of-flight mass spectrometer (Synapt HD MS, Waters, Wilmslow, Manchester, UK)[71,72]. IDH1 was buffer exchanged using a Micro Bio-Spin™ 6 column (Bio-Rad 732-6221) into the water before diluting with ammonium acetate. Protein concentrations were measured using a Nanodrop spectrometer (Thermo Scientific™ NanoDrop™ One). Twenty microlitres of 50 μM wt or R132H IDH1 with compounds of interest in 200 mM ammonium acetate at pH 7.5 were pipetted in the wells on a sample plate of an automated chip-based nano-electrospray ion source (TriVersa Nanomate, Advion, Ithaka, NY, USA). The solution was then sprayed through a nozzle in a Nanomate chip at a spray voltage of 1.8–2.0 kV (spray backing gas pressure 0.8–1.0 psi, inlet pressure 3.8 mbar). The sample and extractor cone voltages were kept at 100 and 5.2 V, respectively, and no in-source dissociation of the protein dimers was observed at these voltages. For promoting collision-induced dissociation of the substrate from its complex with IDH1, a higher cone voltage (200 V) was used. The rest of the voltages and pressures in the mass spectrometer were adjusted for the best resolution and minimum fragmentation. Data collection and analysis were carried out using the Waters MassLynx software. Mass spectra were calibrated externally using the CsI solution to produce CsI cluster ions.

**Cellular studies.** LN18 glioblastoma cells stably overexpressing R132H IDH1(as reported in ref. [43]) were cultured in DMEM (Custom Gibco non-GMP formulation media, 500 mL, Ref. ME19232L1) supplemented with CaCl$_2$ (1.8 mM), sodium pyruvate (1 mM), glucose (25 mM), 10% foetal bovine serum (FBS), 1% Glutamax and MgSO$_4$ (0.8 mM). The MgSO$_4$ concentration was reduced over 3 weeks to 0.05 mM, to enable cell viability at low Mg$^{2+}$ levels. For the inhibitor studies, the LN18 cells were cultured in 6 cm dishes in DMEM with 0.05, 0.1, 0.25, 0.5, 1, 2.5 and 10 mM MgSO$_4$ until they reached 80% confluency, before treating with inhibitor or DMSO control (<1%) for 19 h and harvested. For cell harvesting, the medium was removed and cells were washed twice with PBS (residual PBS was removed), and the cell layer was snap frozen with liquid nitrogen. The cell layer was scraped and metabolites extracted in 200 μL of 80% MeOH/20% H$_2$O. Cell extracts were centrifuged at 18,000 × *g* for 25 min at 4 °C. The double-stranded DNA (dsDNA) concentration in each sample was measured by nanodrop for normalisation after filtration. The supernatant was passed through centrifugal filters (Amicon Ultra Centrifugal Filters, 10 kDa Ultracel, 0.5 mL) by centrifugation at 18,000 × *g* for 25 min at 4 °C. The filtrate of each experimental sample was diluted (based on the lowest dsDNA concentration obtained) to a final volume of 100 μL in 80% MeOH/20% H$_2$O such that the dsDNA concentration of all samples was the same. The % inhibition in cellular studies is calculated by (1 − 2HG levels in treatment/2HG levels in DMSO control) × 100%.

**Measurement of intracellular 2HG levels by AEC-MS.** Samples were analysed by anion-exchange chromatography with MS using a Thermo Scientific ICS-5000+ ion chromatography (IC) system coupled directly to a Q-Exactive Hybrid Quadrupole-Orbitrap mass spectrometer with a HESI II electrospray ionisation source (Thermo Fisher Scientific, San Jose, CA, USA). The electrolytic anion eluent generator (KOH) was programmed to produce an $OH^-$ gradient (over a 37 min gradient and a post-column inline electrolytic suppressor neutralised the $OH^-$ ion content from the post-column eluent stream prior to MS analysis (Thermo Scientific Dionex AERS 500). A 10 μL partial loop injection was used for all analyses and the chromatographic separation was performed using a Thermo Scientific Dionex IonPac AS11-HC $2 \times 250$ mm, 4 μm particle size column. The IC flow rate was 0.250 mL min$^{-1}$. The total run time was 37 min and the hydroxide ion gradient comprised: 0 min, 0 mM; 1 min, 0 mM; 15 min, 60 mM; 25 min, 100 mM; 30 min, 100 mM; 30.1 min, 0 mM; 37 min, 0 mM. MS analysis was performed in negative ion mode using a scan range from $m/z$ 60–900 and resolution set to 70,000. The MS tune file source parameters were set as follows: sheath gas flow 60 mL min$^{-1}$; aux gas flow 20 mL min$^{-1}$; spray voltage 3.6 V; capillary temperature 320 °C; S-lens RF value 70; heater temperature 350 °C. The automatic gain control target was set to 1e6 ions and the max inject time value was 250 ms. The column compartment temperature was kept at 30 °C throughout the experiment. Full scan data were acquired in continuum mode. Raw data files were processed using Progenesis QI (Waters, Elstree, UK). This process included alignment of retention times, peak picking according to natural abundance isotope peak profiles, characterising multiple adducts forms and identification of metabolites using an in-house database of authentic standards. Retention times, accurate mass values, relative isotope abundances and fragmentation patterns were compared between authentic standards and the samples measured. Identifications were accepted when the following criteria were met: <5 p.p.m. difference between measured and theoretical $m/z$ (based on the chemical formula), <30 s differences between authentic standard and analyte retention times (via a historical database) and isotope peak abundance measurements for analytes were >90% matched to the theoretical isotope pattern generated from the chemical formula. Where measured, fragmentation patterns were matched to at least the base peak and two additional peak matches in the MS/MS spectrum to within 12 p.p.m.

**Measurement of intracellular Mg²⁺ levels by CEC-CD.** Intracellular $Mg^{2+}$ levels were analysed by injecting 5 μL of the 100 μL filtered and dsDNA normalised cell extracts (procedure as described above) following treatment of LN18 cells with DMSO control (<1%), AGI-5198 (1 μM) or GSK864 (100 nM) inhibitors in DMEM (Custom Gibco non-GMP formulation media, 500 mL, Ref. ME19232L1) supplemented with $CaCl_2$ (1.8 mM), sodium pyruvate (1 mM), glucose (25 mM), 10% FBS, 1% Glutamax and $MgSO_4$ (0.1, 2.5, and 10 mM).

Samples were analysed by cation-exchange chromatography using a Dionex ICS-5000+ Capillary HPIC system (Dionex, Sunnyvale, CA, USA) with conductivity detection (Thermo Scientific, Waltham, MA, USA), fitted with a Dionex IonPac™ CS19-4μm analytical column (2 ×250 mm²). The method was developed to obtain robust and repeatable separation of $Na^+$, $K^+$, $Ca^{2+}$, $Mg^{2+}$ and abundant biogenic amines found in mammalian cells. The use of CEC for the detection of $Mg^{2+}$ in small biological samples has been previously demonstrated[73]. The column was kept at 30.00 °C and the conductivity detector compartment at 35.00 °C. Partial loop injection (5 μL) was used for all samples. Aqueous methanesulfonic acid was used as mobile phase at flow rate 0.250 mL min$^{-1}$ and gradient with the following steps: 0 min, 5.00 mM; 7 min, 5.00 mM; 15 min, 34.00 mM; 17 min, 65.00 mM, 22 min, 65.00 mM, 22.1 min, 5.00 mM; 29 min, 5.00 mM. A Dionex Cation Electrolytically Regenerated Suppressor 500e (2 mm) was used for ion suppression and was run in external water mode at a flow rate 0.500 mL min$^{-1}$ with 96 mA suppressor current. Calibration solutions of $MgSO_4·7H_2O$ (99.0%, Sigma-Aldrich) in water at 0.025, 0.050, 0.075, 0.100, 0.250, 0.500, 0.750, 1.000 and 2.500 mM were analysed with water blanks between each injection of calibration solution. Prior to the analysis of cell samples, a pooled quality control sample was analysed five times and for every seven cell samples within the analytical sequence. The analyte peak area from conductivity detection of authentic standards was used to construct a calibration curve from which sample analyte concentrations were derived.

**Statistics and reproducibility.** Biochemical assays employed (at least) three technical replicates in a plate format. DSF assays typically employed six doses with three technical replicates for each. ITC, NMR and non-denaturing MS experiments were done in a single replicate on one day; this was reproduced independently on separate days, using two different inhibitors wherever applicable. Three tissue cultures replicates were used for cell studies, and experiments were reproduced on separate days.

**Reporting summary.** Further information on research design is available in the Nature Research Reporting Summary linked to this article.

## Data availability
The source data underlying the graphs in the main figures/table are available in Supplementary Data 1. Other datasets generated and/or analysed during the current study are available from the corresponding author on reasonable request.

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

## Acknowledgements

C.J.S. thanks the Biotechnology and Biological Sciences Research Council (BBSRC), the Wellcome Trust (106244/Z/14/Z) and Cancer Research UK (C8717/A18245) for funding. S.L. thanks the Agency for Science, Technology and Research (A*STAR, Singapore) for a National Science Scholarship. T.J. was supported by the Oxford-GSK-Crick Doctoral Programme in Chemical Biology, EPSRC (EP/R512060/1) and GlaxoSmithKline. I.H. thanks Anne Grete Eidsvig and Kjell Inge Røkkes Charitable Foundation for Education for an Aker Scholarship.

## Author contributions

S.L., M.I.A. and C.J.S. designed the experiments. S.L. conducted biochemical assays, DSF, ITC and NMR experiments. V.M. and S.L. conducted and analysed non-denaturing MS experiments. T.J., J.W.-T. and S.L. conducted cell studies. J.W.-T. measured 2HG by AEC-MS. I.H. measured Mg$^{2+}$ by CEC-CD. M.I.A. initiated experimental work and prepared the recombinant proteins. I.P. prepared the expression vector. X.L. conducted additional biochemical assays, DSF and ITC experiments following paper review. C.J.S., J.S.O.M. and T.C.-H. supervised the work. S.L. prepared the figures and made an initial draft of the manuscript, which S.L. and C.J.S. subsequently wrote. All authors reviewed the manuscript.

## Competing interests

The authors declare no competing interests.
