## [Transparent Peer Review File · Communications Biology]

Reviewers' comments:

Reviewer #1 (Remarks to the Author):

The work by Liu et al. investigates the effect of divalent metal ions on catalysis and inhibition of wildtype and a cancer linked variant of the enzyme IDH. IDH enzymes catalyze the oxidative decarboxylation of isocitrate to 2OG in the TCA cycle. Mutations in the IDH gene are linked to cancer. The enzyme forms dimers and most of the inhibitors developed for these enzymes bind at an allosteric site located at the dimer interface. The authors propose that these findings are related to the important role of divalent metal ions in substrate and inhibitor binding. The authors studied the effects of Mg²⁺ and Mn²⁺ ions on substrate binding, showing that the presence of these divalent metal ions enhances the substrate binding to IDH2 (wt and R132H variant). Using a combination of ITC and mass spectrometry, they showed that while isocitrate binds to IDH1 as a Mg²⁺ complex, 2-oxoglutarate does not form a strong complex with Mg²⁺ when binding to the variant. Furthermore, they showed that the mode of action of R132H IDH1 inhibitors that bind to the dimer interface involves weakening the binding of Mg²⁺ complexes. Overall, the work is of great interest as it investigates an enzyme that is part of a key process (TCA cycle) and a variant of IDH 1 that is related to cancer. While many inhibitors have been reported yet, the role of metal ions in the binding of substrate and inhibition is not well understood. The role of metal ions is very important, however, for many biological systems it has not been investigated in detail. Previous studies have also suggested that the lack of understanding the role of metals leads to large discrepancies between biochemical (in vitro) and cell (in vivo) experiments. Therefore, the present study contributes significantly to a better understanding of the influence of metal ions. The scientific quality of the work is high. However, a major concern in the focus on divalent metal ions, although, monovalent metal ions are equally important.

Major points:

1. Why was the effect of Mn²⁺ ions on isocitrate oxidation and 2OG reduction studied? Mn ions do not play a significant role inside the cell; thus, it would be necessary to explain why this specific metal was investigated in addition to Mg²⁺, especially in the first section of the results.
2. The effects of magnesium and manganese ions on kinetic parameters were studied in the absence of NaCl or KCl. Inside the cell, the concentration of monovalent ions is quite high (140 mM for K⁺ and 10 mM for Na⁺ in mammalian cells according to Pechlaner and Sigel 2011). Thus, the effects of the divalent metal ions could be a result of a change in ionic strength and are not representative of the biological relevant conditions. Experiments showing that the observed kinetic parameters in presence of 150 mM monovalent ions show a similar effect are necessary for these findings to be relevant.
3. The binding of isocitrate to wt IDH1 (ITC experiments) should also be re-evaluated in presence of 150 mM monovalent ions. In general, no differences between NaCl and KCl are observed, therefore, using one of the should be sufficient.
4. For the studies on R132H IDH1 inhibition, one control experiment at 150 mM monovalent ions should be provided, to show that the metal ion effect at physiological ionic strength is present.

Minor points:

Line 29: The abbreviation 2OG has not been introduced previously.

Line 110: The term "redox neutral" should be changed to "non-redox"

Figure 6: Error bars are hard to see, so they should be shown in black.

Figure 7: The text is not aligned. Why are there two arrows between the penultimate and last reactants in a) and b)?

Reviewer #2 (Remarks to the Author):

The manuscript "Roles Of Metal Ions In Catalysis And Allosteric Inhibition Of Wildtype And Oncogenic

Variants Of Human Isocitrate Dehydrogenase (IDH1)" by Liu et al. rigorously reveals an additional and critical layer of the mechanistic basis of inhibition by IDH1 lead compounds. Much hope rests on the current crop of mutant-selective IDH1 inhibitors for the treatment of IDH mutant cancers. In order to continue development of these inhibitors, to make them more potent and more selective and less prone to resistance, a better understanding of how the current crop of inhibitors work would be helpful. For example, the mechanistic basis for the selectivity of mutant-selective IDH1 inhibitors is not yet well understood. This study reveals for the first time, using a host of biophysical techniques, that the enzyme, catalyzing the forward reaction, preferentially binds the isocitrate substrate and cofactor magnesium simultaneously, and not in a sequential manner, but rather as a chelate complex. This observation in itself is terribly interesting but does not show the mechanistic basis for the selectivity of mutant-selective IDH1 inhibitors. This study further suggests with significant orthogonal evidence that the enzyme, catalyzing the reverse reaction, preferentially binds 2OG substrate and cofactor sequentially with magnesium presumably binding first or leaving last. Inhibitors therefore can compete with magnesium binding, but apparently are impotent against chelates. The implications could be significant; are assays being developed optimally; should we pay more attention to magnesium binding as opposed to the overall reaction or substeps; are there other enzymes or proteins that prefer binding chelates; is this an evolutionarily conserved mechanism; are there other inhibitors that can differentiate between chelates and nonchelates; are divalent metal ions per se novel therapeutic targets. Overall, this well-written manuscript is an important contribution to the field of mutant IDH diseases (glioma and AML for example) as well as the large field of metal-mediated reactions and binding events as it provides high-resolution details into the mechanism of reaction and inhibition.

Reviewer #3 (Remarks to the Author):

The findings provide insight into the mechanism of R132H IDH1 inhibition and potential ways to improve the targeting of this mutation of such important enzyme.

Their results have implications for the mechanism by which selective inhibition for R132H over wt IDH1 is achieved for AG-120, GSK864 and, by implication, related allosteric inhibitors.

One interesting finding is the fact that cellular assays had a way to compensate for Mg. Is Mg homeostasis something that would also occur in vivo? In other words what is the concentration of Mg in these cancer cells and how is that playing a role in the model that authors put forth?

Responses to the reviewers' comments

We thank all 3 reviewers for their very positive comments and careful reading of the manuscript. We are very pleased that they found the work of 'great interest', 'terribly interesting' and that it 'contributes significantly'. We have addressed all the minor points said by the reviewers and have carried out experiments to address the major points raised by reviewer 1.

Reviewer 1

Major points:

Point 1. *Why was the effect of Mn²⁺ ions on isocitrate oxidation and 2OG reduction studied? Mn ions do not play a significant role inside the cell; thus, it would be necessary to explain why this specific metal was investigated in addition to Mg²⁺, especially in the first section of the results.*

Mn²⁺ ions are essential for humans (see e.g. Front BioSci, 2018, 23, 1655-1679). Mn²⁺ ions often, but not always, can substitute for Mg²⁺ ions. We have made this clear in the revised results section and added 3 references concerning Mn²⁺ ions.

Point 2. *The effects of magnesium and manganese ions on kinetic parameters were studied in the absence of NaCl or KCl. Inside the cell, the concentration of monovalent ions is quite high (140 mM for K⁺ and 10 mM for Na⁺ in mammalian cells according to Pechlaner and Sigel 2011). Thus, the effects of the divalent metal ions could be a result of a change in ionic strength and are not representative of the biological relevant conditions. Experiments showing that the observed kinetic parameters in presence of 150 mM monovalent ions show a similar effect are necessary for these findings to be relevant.*

Thank you for this suggestion – the possibilities of ionic strength having an effect is a really good point. In fact we already had some data to investigate this point but have carried out further studies, as now included as **Figure S2d, S2m**. The results, show only small, if any, changes in the pressure of 150mM NaCl.

Point 3. *The binding of isocitrate to wt IDH1 (ITC experiments) should also be re-evaluated in presence of 150 mM monovalent ions. In general, no differences between NaCl and KCl are observed, therefore, using one of the should be sufficient.*

We have obtained new ITC and DSF data in the presence of 150 mM NaCl – again this is consistent with our data indicating that for the Mg²⁺ isocitrate complex is the true IDH1 substitute. See addition as **Figure S7a** (ITC) and **S8a** (DSF).

Point 4. *For the studies on R132H IDH1 inhibition, one control experiment at 150 mM monovalent ions should be provided, to show that the metal ion effect at physiological ionic strength is present.*

We have carried out the requested control (again a very good point) by obtaining IC₅₀ values for one inhibitor at varied NaCl concentrations (added as **Figure 6b, 6c**).

Minor points:

*Line 29: The abbreviation 2OG has not been introduced previously.
2OG has been defined in full*

*Line 110: The term “redox neutral” should be changed to “non-redox”
‘redox neutral’ has been changed to ‘non-redox’*

Figure 6. Error bars are hard to see, so they should be shown in black.

*The error bars are now in black (in the journal style – also for **Figure 2**)*

Figure 7. The text is not aligned. Why are there two arrows between the penultimate and last reactants in a) and b)?

We have corrected the text. The 2 arrows in the final steps of indicate that more than one step is involved.

Reviewer 2

We thank the reviewer for their positive comments and appreciation of the broad importance of the results.

Reviewer 3

One interesting finding is the fact that cellular assays had a way to compensate for Mg. Is Mg homeostasis something that would also occur in vivo? In other words what is the concentration of Mg in these cancer cells and how is that playing a role in the model that authors put forth?

*We thank the reviewer for their support of the work. We have added clarification on the issue of Mg²⁺ homeostasis. We agree that it is of interest to study Mg²⁺ concentrations in actual IDH mutant bearing cells, though this is out the scope of our *in vitro* work. Please note we have included references (41, 55) concerning the free and total Mg²⁺ concentrations in brain tumours.*

REVIEWERS' COMMENTS:

Reviewer #1 (Remarks to the Author):

The additional experiments provide more convincing evidence for the observations and conclusions drawn from the data. I think that the authors have addressed all concerns appropriately and the additional data strengthen the manuscript.